Corrected: Publisher correction

# Sensory overamplification in layer 5 auditory corticofugal projection neurons following cochlear nerve synaptic damage

Meenakshi M. Asokan[1,2], Ross S. Williamson[1,3], Kenneth E. Hancock[1,3] & Daniel B. Polley[1,2,3]

Layer 5 (L5) cortical projection neurons innervate far-ranging brain areas to coordinate integrative sensory processing and adaptive behaviors. Here, we characterize a plasticity in L5 auditory cortex (ACtx) neurons that innervate the inferior colliculus (IC), thalamus, lateral amygdala and striatum. We track daily changes in sound processing using chronic widefield calcium imaging of L5 axon terminals on the dorsal cap of the IC in awake, adult mice. Sound level growth functions at the level of the auditory nerve and corticocollicular axon terminals are both strongly depressed hours after noise-induced damage of cochlear afferent synapses. Corticocollicular response gain rebounded above baseline levels by the following day and remained elevated for several weeks despite a persistent reduction in auditory nerve input. Sustained potentiation of excitatory ACtx projection neurons that innervate multiple limbic and subcortical auditory centers may underlie hyperexcitability and aberrant functional coupling of distributed brain networks in tinnitus and hyperacusis.

[1] Eaton-Peabody Laboratories, Massachusetts Eye and Ear Infirmary, Boston, MA 02114, USA. [2] Division of Medical Sciences, Harvard University, Boston, MA 02114, USA. [3] Department of Otolaryngology, Harvard Medical School, Boston, MA 02114, USA. Correspondence and requests for materials should be addressed to M.M.A. (email: masokan@g.harvard.edu)

The auditory system employs a variety of gain control mechanisms to encode fluctuations in acoustic signal energies that can vary by over a million-million fold (120 dB). Auditory gain control places a premium on speed, often activating within tens or hundreds of milliseconds following sudden changes in sound level to protect the ear from over-exposure and adjust the dynamic range of neural coding[1,2]. In addition to these fast-acting gain control systems, central auditory neurons also exhibit slower gain control systems that increase neural excitability following peripheral afferent damage over time scales ranging from days to months[3].

Descending centrifugal projections may play an important role in auditory gain control. For example, brainstem efferent neurons change the acoustic impedance of the middle ear and dampen excitability of cochlear sound transduction to protect the inner ear and normalize activity levels in the auditory nerve[4]. The largest descending auditory pathway arises from neurons in the deep layers of the auditory cortex (ACtx) that innervate nearly all levels of subcortical auditory processing as well as many structures outside of the classical auditory pathway such as the lateral amygdala and striatum[5,6]. Less is known about how corticofugal neurons support various forms of central gain control[7,8]. Although non-selective lesions, inactivation or stimulation of ACtx neurons can have striking effects on subcortical auditory responses, the effects are often heterogeneous, with neurons in the same brain region showing diverse forms of modulation[9–14].

Corticofugal neurons themselves are not a singular cell type, but rather comprise a diverse set of projection neurons with distinct local inputs, subcortical targets, intrinsic properties and synaptic properties[15–20]. Traditional approaches to characterize the effects of cortical feedback on subcortical sound processing and plasticity through cooling, pharmacological silencing or microstimulation indiscriminately manipulate multiple types of corticofugal neurons as well as interneurons, intracortical projection neurons or even axons of passage. This technical limitation may explain why the subcortical effects of ACtx manipulations are often heterogeneous and has generally hampered progress in understanding how corticofugal neurons contribute to auditory processing and gain control. Recent efforts have begun to circumvent these limitations by using approaches to lesion[21,22], rewire[23] or optogenetically activate and silence select classes of auditory projection neurons[24–27]. While paradigms to artificially manipulate the activity of corticofugal pathways have their appeal, there is also a need to monitor the activity of select classes of corticofugal neurons and describe how naturally occurring plasticity in their auditory response profiles support central gain adjustments across a variety of time scales. To this end, we adapt a widefield calcium imaging approach to track daily changes in sound processing from the axons of ACtx neurons that project to the inferior colliculus (IC)[28,29]. We describe rapid adjustments and persistent potentiation in corticocollicular (CCol) response gain that compensates for a loss of peripheral input following noise-induced cochlear synaptic damage.

## Results

**Distributed targets of ACtx corticocollicular neurons.** Layer 5 (L5) pyramidal cells are the canonical broadcast neurons of the cortex, with far-ranging projections throughout the neocortex, striatum, amygdala, thalamus, midbrain and brainstem[19,30]. Dual retrograde tracer studies have emphasized that ACtx L5 projections to downstream targets are anatomically separate, such that L5 neurons that project to the inferior colliculus (CCol) are largely separate from those that project to the lateral amygdala, contralateral cortex and so forth[31,32] (though prior work has identified a small fraction of double-labeled cells that project both

to the IC and striatum[33] or both to the IC and brainstem[34]). Interpreting the findings of dual retrograde tracer studies is challenging, as there is a risk of underestimating the true prevalence of projection neurons that innervate multiple downstream targets. Because tracer injections fill only a fraction of the target nucleus, the entirety of an axon projection zone (or portions thereof) could be missed by one of the tracers, leading to false negatives. Secondly, dual tracer studies can only identify divergence to a maximum of two downstream structures leaving unanswered the possibility that cortical neurons could broadly innervate multiple targets[20].

While ground truth estimates of projection diversity will ultimately require whole brain reconstructions of individual cells, we used an intersectional virus strategy to determine whether the axons of at least some CCol projection neurons also innervate other structures. This was accomplished by first injecting a canine adenovirus 2 (CAV2), which offers a strong bias for retrograde infection into the IC ($n = 2$ mice)[35–37]. With cre-recombinase expressed in neurons that project to the IC, we then injected a cre-dependent virus into the ipsilateral ACtx to express a fluorescent marker throughout CCol neuron axon fields. We observed labeled L5 cell bodies and strong terminal labeling in the external and dorsal cortex of the IC, as expected (Fig. 1a, top row). Interestingly, we also observed terminal labeling of CCol axon collaterals in the dorsal subdivision of the medial geniculate body (MGB) (Fig. 1a, middle row), caudal regions of the dorsal striatum and the lateral amygdala (Fig. 1a, bottom row). Although well known that L5 neurons of ACtx project to each of these targets individually, the intersectional viral labeling strategy used here suggested that at least some CCol neurons have multiple far-ranging projections to other structures throughout the ipsilateral forebrain.

**Visualizing sound-evoked activity from CCol neurons.** Having established that at least some CCol neurons comprise a broader, widespread corticofugal projection that also innervates the auditory thalamus, dorsal striatum and lateral amygdala, we next developed a calcium imaging approach to monitor daily changes in their activity levels (Fig. 1b). We reasoned that 2-photon imaging of L5 CCol cell bodies would be quite challenging on account of the depth from the surface and prominent apical dendrites. Instead, we adapted a protocol to express the genetically encoded calcium indicator GCaMP6s in the ACtx and then image sound-evoked responses from CCol axons on the dorsal surface of the brain, atop the IC (Fig. 1c)[28,29]. By implanting custom head-restraint hardware and a cranial window[38] over the dorsal cap of the IC, we were able to perform daily widefield epifluorescence imaging of CCol axon population activity in awake mice (Fig. 2a). We observed that CCol axon response amplitude increased monotonically with sound level, as measured from the peak fractional change in GCaMP6s amplitude evoked by a brief (50 ms) broadband noise burst (Fig. 2b, c). By contrast, sound level growth functions were fairly flat when signals were measured from more caudal locations within the imaging window (Fig. 2c, blue line), demonstrating that responses could not be attributed to non-specific changes in time-locked intrinsic signals or autofluorescence measured from brain areas without GCaMP6s expression.

To assess the stability of CCol response growth functions over time, we repeated the imaging experiment for seven consecutive days in each mouse ($n = 5$). Qualitatively, we observed a fairly consistent monotonic growth in CCol response amplitude, as shown in a representative example mouse (Fig. 2d, top row). Gain describes a change in output per unit change in input (e.g., CCol response amplitude per unit increase in dB sound pressure level

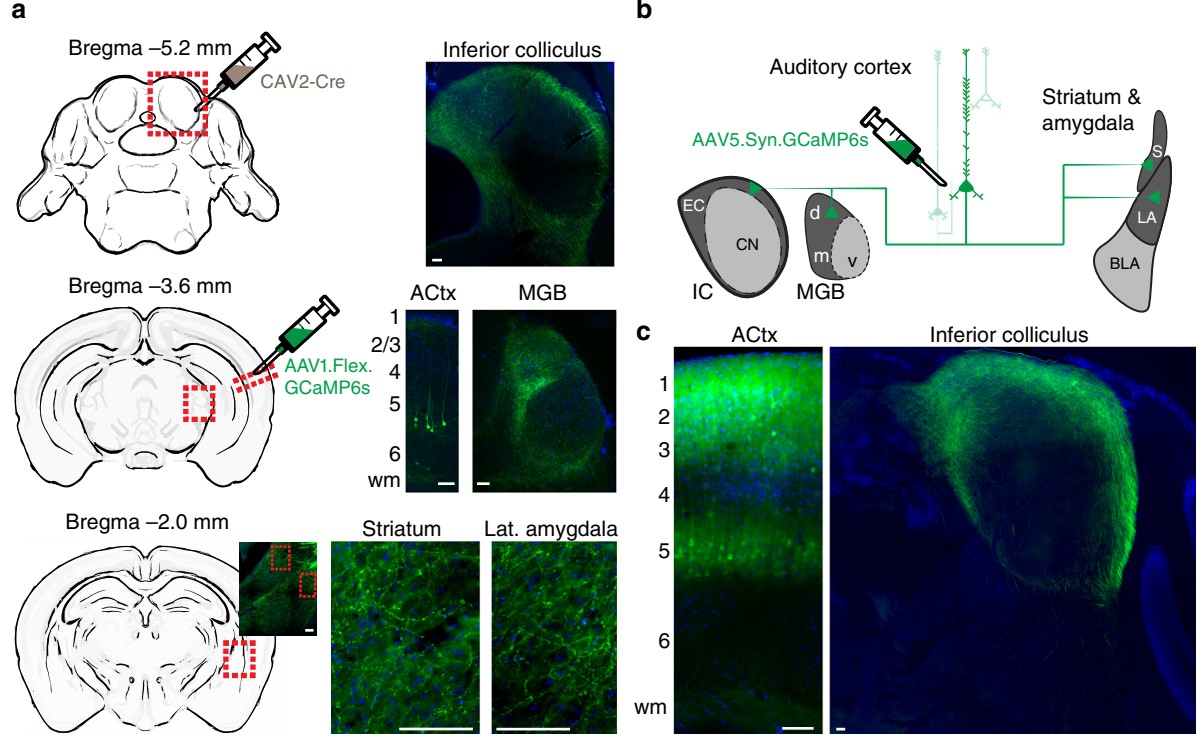

**Fig. 1** Auditory corticofugal neurons that innervate the inferior colliculus have other widespread targets throughout the forebrain. **a** A canine adenovirus vector with efficient retrograde transport (CAV2) was injected into the IC to express cre-recombinase in neurons that project into the injection zone. A cre-dependent AAV was then injected into the ipsilateral ACtx to express a fluorescent marker throughout the entire axon field of CCol neurons. Photomicrographs show the expected labeling of layer 5 ACtx neurons and their IC axon terminals, with additional strong axon labeling in the dorsal nucleus of the medial geniculate body, lateral amygdala and posterior regions of the dorsal striatum. wm white matter. **b** Schematic of virus strategy used for in vivo Ca$^{2+}$ imaging in corticofugal axons. EC and CN are external cortex and central nucleus of the IC, respectively. MGB subdivisions d, v and m are dorsal, ventral and medial, respectively. LA and BLA are lateral and basolateral amygdala, respectively. S is striatum. **c** Strong labeling of L5 pyramidal neuron cell bodies, apical dendrites and CCol axon terminals are observed approximately 5 weeks after injection of the GCaMP6s virus in ACtx. All scale bars = 0.1 mm

(SPL)). We quantified changes in CCol response gain across the linear portion of the sound level growth function (40–80 dB SPL), as the transformation between the mean CCol response growth measured during the first two imaging sessions ($r_{baseline}$) to the CCol response growth measured on any given day ($r_{day}$) according to the formula $r_{day} = m \times r_{baseline} + c$. With this approach, the slope of the linear fit ($m$) describes the multiplicative ($m > 1$) or divisive ($m < 1$) change in response growth on any given imaging session with respect to the baseline period (Fig. 2d, bottom row). In the absence of any explicit perturbation, we observed that CCol gain changes over a 7-day imaging period were minimal (two-way repeated measures analysis of variance (ANOVA), main effect for imaging session, $F = 1.15$, $p = 0.36$; sound level × session interaction term, $F = 1.34$, $p = 0.12$, $n = 5$, Fig. 2e).

**Moderate noise exposure damages cochlear afferent synapses.** Having established that CCol response gain is relatively stable from one day to the next in a control condition, we next addressed whether and how corticofugal outputs from the ACtx increase response gain to compensate for a loss in peripheral input. Isolating dynamics in central gain is challenging with protocols that induce widespread cochlear damage, because the loss of outer hair cell-based amplification introduces complex changes in cochlear tuning that are inextricable from changes arising through central plasticity. For this reason, central gain dynamics in intact preparations are most readily studied with

hearing loss protocols that selectively eliminate cochlear afferent neurons in the spiral ganglion or their peripheral synapses onto inner hair cells without inducing permanent changes to cochlear transduction and amplification mechanisms.

We implemented a protocol to track changes in the auditory brainstem response (ABR) and a non-invasive measure of outer hair cell function, the distortion product otoacoustic emission (DPOAE), following noise exposure that was calibrated to damage cochlear afferent synapses at the high-frequency base of the cochlea without causing permanent damage to cochlear hair cells[39]. Following baseline measurements, mice were exposed to a continuous band of octave-wide noise (8–16 kHz at 100 dB SPL) for 2 h (Fig. 3a). As described in many previous studies[40], this moderate-intensity noise induced a temporary shift in DPOAE and ABR thresholds measured 24 h after noise exposure before returning to baseline levels when tested again, several weeks later (repeated measures ANOVA, $F > 30$, $p < 0.00001$ for both DPOAE and ABR threshold shift at 24 h versus 2 weeks, Fig. 3b, c, respectively).

Wave 1 of the ABR is generated by Type-I spiral ganglion neurons, where the amplitude is proportional to the number of their intact synapses onto inner hair cells[39–41]. Prior work has demonstrated that a reduced amplitude of ABR wave 1 can reflect a hidden degeneration of primary cochlear afferents that is not detected by standard measurements of DPOAE and ABR threshold shift[40]. We confirmed this observation in our data; 24 h following noise exposure, ABR wave 1 amplitude was reduced at test frequencies ranging from 11.3 to 32 kHz

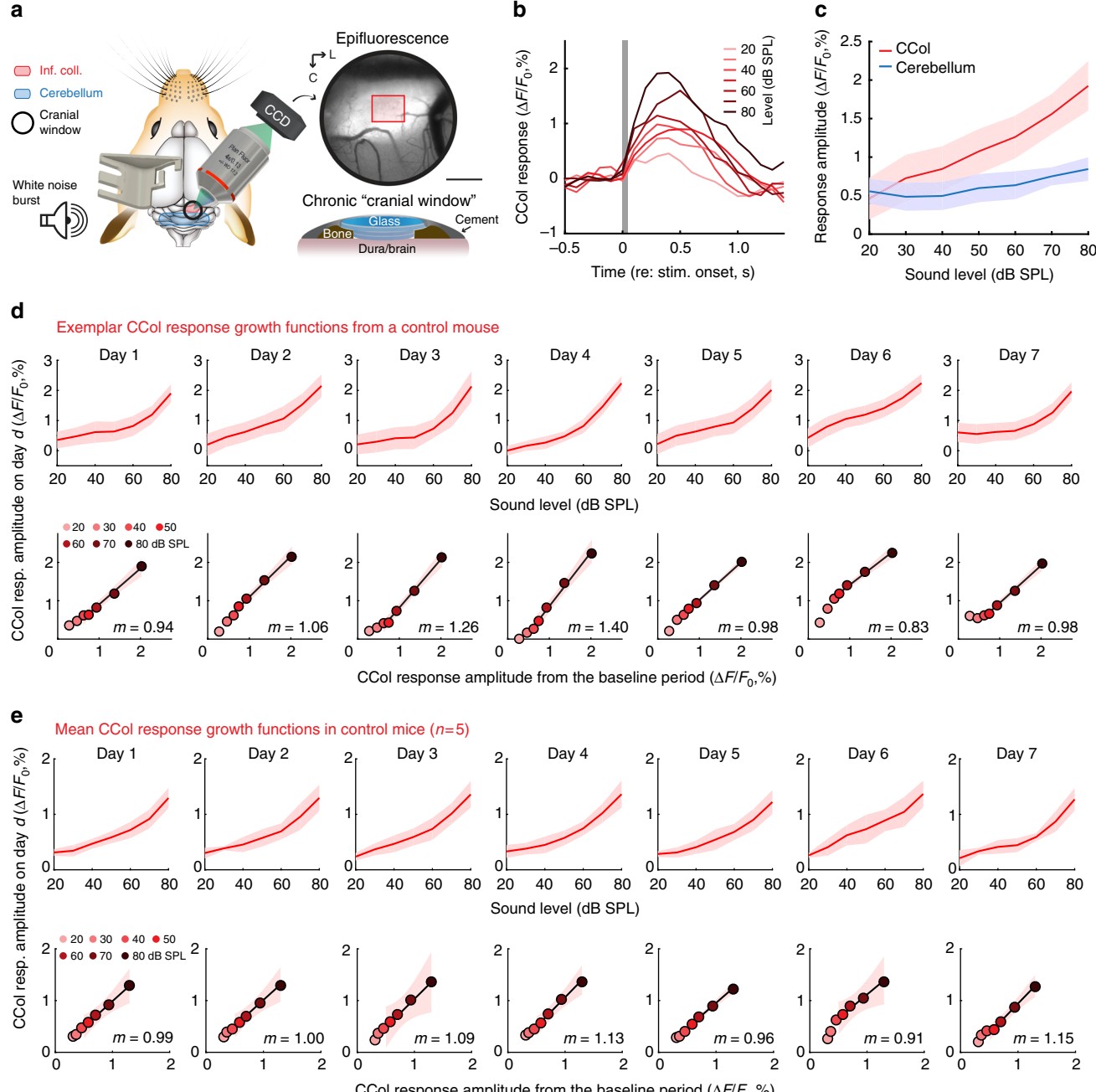

**Fig. 2** Sound-evoked corticocollicular axon response increases monotonically with sound level and remains stable over 1 week of imaging. **a** A chronic preparation for epifluorescence imaging of GCaMP6s in CCol axons via a cranial window (scale bars = 0.5 mm) in awake, head-fixed mice. Red rectangle denotes region of interest for CCol imaging. L is lateral and C is caudal. Mouse schematics in Figs. 2a, 3a, and 4a are adapted with permission from Aronoff et al., 2010[69]. All rights reserved. These images are not covered under the CC BY license for this article. Scientific data and content are original. **b** Time course of mean fractional change in the CCol response amplitude evoked by a 50 ms white noise burst from a single imaging session. Gray box denotes stimulus timing and duration. **c** The monotonic growth of CCol peak response amplitude falls off steeply when the region of interest is shifted away from the IC. Data represent mean ± SEM. **d** Top: CCol response growth functions from a single mouse across seven daily imaging sessions. Data represent mean ± SEM. Bottom: Scatterplots depict the mean CCol response amplitude (x-axis) at each sound level measured from the first two imaging sessions (defined as baseline) against the CCol response amplitude (y-axis) measured on the day specified. The slope (m) of the linear fit provides an estimate of daily changes in response gain, where $m = 1$ indicates a matched response growth relative to baseline, $m < 1$ indicates a divisive flattening of the growth function and $m > 1$ indicates a multiplicative enhancement relative to baseline. Shading represents the 95% confidence interval of the fit. **e** As per **d**, averaged across all control mice (n = 5)

(repeated measures ANOVA, baseline vs day 1, $F > 12$, $p < 0.005$ for 11.3–32 kHz tones; Fig. 3d gray vs orange). When measured again 2 weeks after noise exposure, a full recovery was observed at low- and mid-frequencies, yet wave 1 amplitude remained significantly reduced at 22.6 and 32 kHz (repeated measures

ANOVA, baseline vs 2 weeks, $F < 2.1$, $p > 0.05$ for 8–16 kHz; $F > 9$, $p < 0.005$ for 22.6 and 32 kHz, Fig. 3d, gray vs red). To confirm that reduced ABR wave 1 amplitude was associated with a loss of cochlear afferent synapses, we quantified immunolabeling of auditory nerve synapses onto inner hair cells in the high-

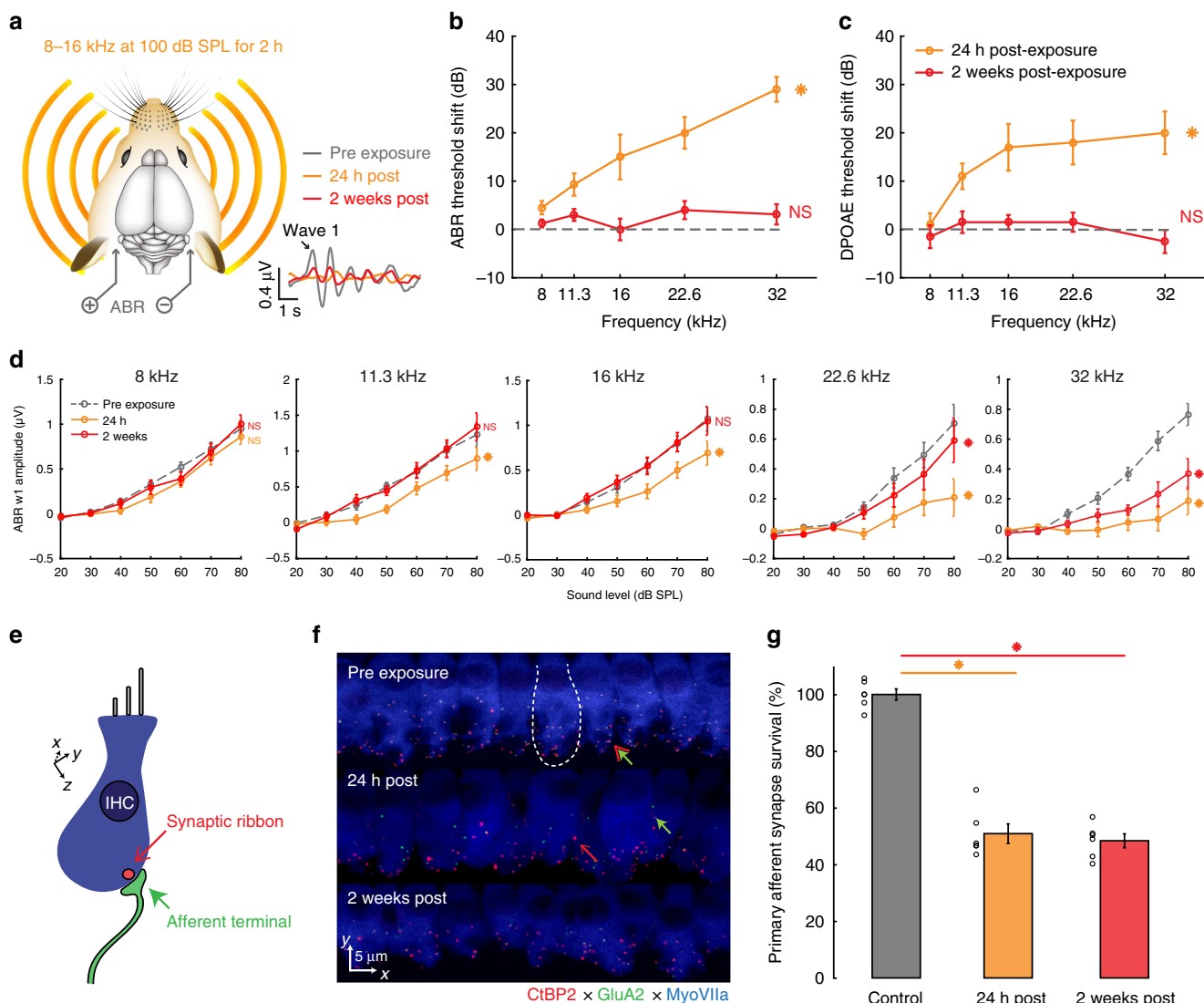

**Fig. 3** Moderate intensity noise exposure induces a temporary shift in cochlear and brainstem response thresholds but a permanent loss of auditory nerve afferent fibers. **a** Schematic of noise exposure and auditory brainstem response (ABR) measurement protocols. Example ABR waveforms evoked with a 32 kHz tone bursts before, 24 h after and 2 weeks after noise exposure. **b**, **c** Elevations in ABR and distortion product otoacoustic emission (DPOAE) thresholds (**b** and **c**, respectively) are observed 1 day following noise exposure (orange) but have returned to baseline 2 weeks following noise exposure (red). **d** ABR wave 1 (w1) growth functions. NS indicates no significant difference with pre-exposure. Asterisk indicates significant main effect for ABR amplitude between pre-exposure and post-exposure. Data represent mean ± SEM, $n = 10$ mice in pre-exposure and 24 h post conditions, $n = 8$ mice for 2 weeks post. **e**, **f** Schematic (**e**) and actual (**f**) visualizations of cochlear nerve afferent synapses on inner hair cells. Red (open) and green (closed) arrowheads depict orphaned presynaptic ribbons and postsynaptic GluA2 receptor patches, respectively. Combined red (open) and green (closed) arrowhead identifies primary afferent cochlear synapses as appositions of the CtBP2 and GluA2 pre- and postsynaptic markers, respectively. Dashed white line depicts the boundary of a single inner hair cell. **g** Quantification of cochlear afferent synapses in control mice, 24 h and 2 weeks following noise exposure. Synaptic counts are expressed as percent survival by comparison to normative standards from age- and strain-matched mice[41,42]. Asterisk indicates significant difference with an unpaired *t*-test after correcting for multiple comparisons. Synaptic counts were made from 20.77 – 21.73 individual inner hair cells at a fixed position in the cochlear frequency map between the 20 and 30 kHz region in each ear, 3 ears per group and 2 cochlear sections per ear

frequency base of the cochlea (Fig. 3e, f). We found that approximately 50% of cochlear afferent synapses were eliminated when measured 24 h after noise exposure or 2 weeks following noise exposure, as reported previously (synaptic counts were made from 20.77 ± 0.02 to 21.73 ± 0.63 inner hair cells per ear in all groups, 3 ears per group, unpaired *t*-tests, $p < 1 \times 10^{-8}$ for both control vs 24 h and control vs 2 weeks after correcting for multiple comparisons; Fig. 3g)[42].

**CCol gain potentiation despite reduced auditory nerve input**. To contrast changes in sound level growth functions measured

in the auditory nerve and CCol axons following cochlear synaptopathy (Fig. 4a), we tracked the day-to-day changes in wave 1 amplitude and CCol response amplitude evoked by a broadband noise burst before and after moderate noise exposure (Fig. 4b). As predicted from cochlear function testing with tone bursts, wave 1 growth functions were depressed following noise exposure and did not recover to baseline levels (Fig. 4c). Although CCol gain was pegged to wave 1 in the first hours following noise exposure, we observed that the gain was increased above baseline levels by day 2 (D2) (2-way repeated measures ANOVA, main effect for imaging session, $F > 9$, $p < 1 \times 10^{-8}$;

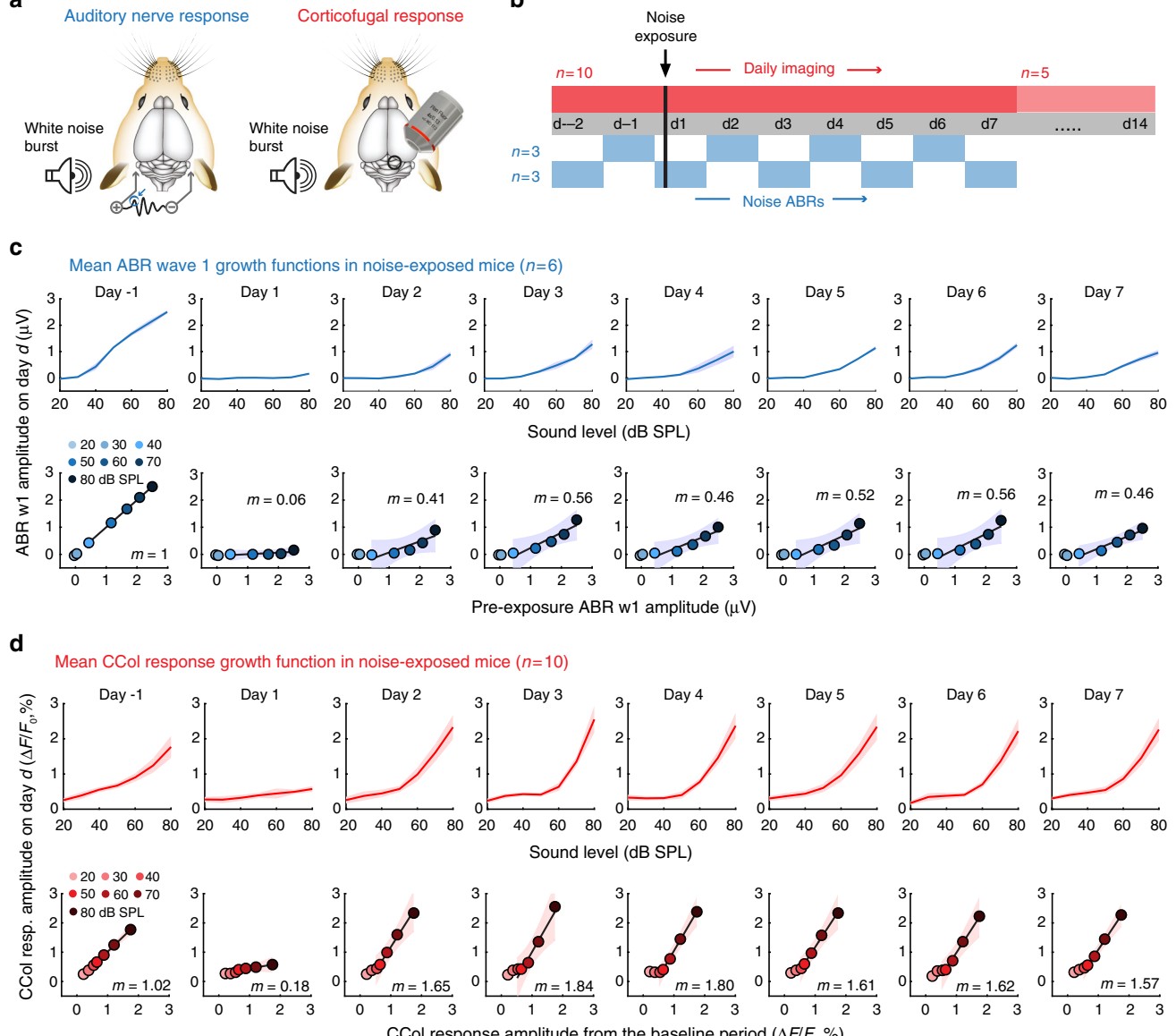

**Fig. 4** Opposing changes in auditory nerve and corticocollicular response growth functions following cochlear synaptopathy. **a** Auditory nerve growth functions were measured under anesthesia every other day according to the change in ABR wave 1 (blue circle) amplitude to white noise bursts of varying level ($n = 6$). CCol response growth functions were measured daily in a separate cohort of awake mice ($n = 10$) also using white noise bursts, per previous figures. **b** ABR wave 1 and CCol responses were both measured for 2 days (d) prior to moderate noise exposure and for 7 days following noise exposure. In a subset of noise-exposed mice ($n = 5$), CCol imaging was extended for an additional week after noise exposure. **c, d** As per Fig. 2e, ABR wave 1 (**c**) and CCol response (**d**) growth functions (top rows) and scatterplots of linear fits for baseline vs post-exposure growth functions (bottom rows) are provided for all mice. Data represent mean ± SEM. Linear fits of the five highest sound levels are illustrated by the solid black line with corresponding slope ($m$) and 95% confidence interval (blue and red shading)

imaging session × sound level interaction term, $F > 7$, $p < 0.05$, $n = 10$, Fig. 4d).

A side-by-side comparison of daily changes in noise-evoked CCol response gain, ABR wave 1 amplitude and wave 1 threshold highlights the distinct regulation of each signal. The moderate-intensity noise exposure protocol reversibly elevates ABR thresholds for 1–2 days, based on transient changes in cochlear biomechanics (Fig. 5a). ABR responses at threshold reflect outer hair cell integrity and activation of low-threshold auditory nerve fibers. The moderate-intensity noise exposure protocol used here primarily eliminates synapses from higher-threshold auditory nerve fibers onto inner hair cells[39]. Therefore, a substantial loss of auditory nerve afferent fibers can hide behind normal ABR thresholds, but can be reliably revealed by measuring the growth of ABR wave 1 amplitude across a range of suprathreshold sound levels[39,42]. We observed a pronounced reduction in ABR wave 1 amplitude hours after noise exposure that reflected the combined loss of auditory nerve synapses and additional transient biomechanical changes that underlie the temporary threshold shift[40]. Suprathreshold response gain in the auditory nerve partially recovered from D1 to D2, as the sources of temporary threshold shift reversed, leaving ~60% reduction in the auditory nerve growth slope through D7 that presumably arose from the loss of approximately 50% loss of high-frequency cochlear afferent synapses (Fig. 5b, blue line).

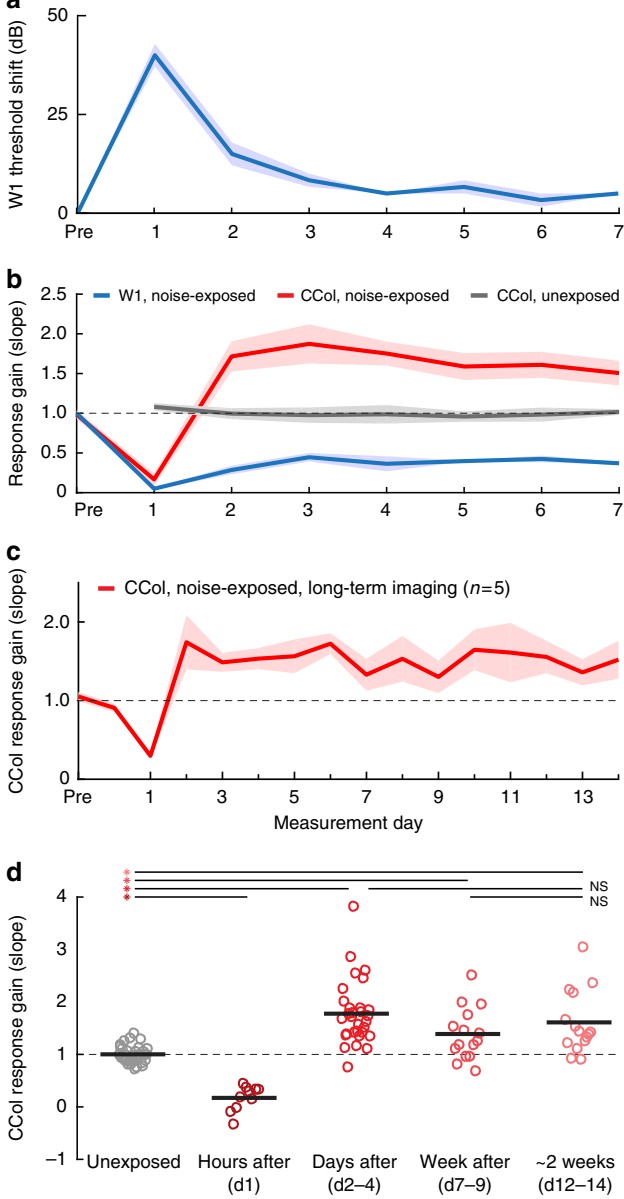

**Fig. 5** ABR threshold recovery belies ongoing dynamics in auditory nerve and corticocollicular response gain. **a** Moderate noise exposure induces a temporary shift in the ABR wave 1 threshold to white noise bursts that resolved after 2 days. **b** Daily changes in response gain for CCol measurements in unexposed control (gray, $n = 5$) and noise-exposed (red, $n = 10$) mice are contrasted with daily changes in the response gain of ABR wave 1 in noise-exposed mice (blue, $n = 6$). In all cases, gain is calculated as the slope of the fit line applied to sound level growth functions measured during baseline and subsequent days. **c** Daily changes in CCol response gain over an extended 2-week imaging period in a subset of noise-exposed mice ($n = 5$). For **a–c**, data represent mean ± SEM. **d** Gain estimates from individual imaging sessions in unexposed control mice (gray) are contrasted with gain estimates measured during the first imaging session following noise exposure (hours after), or during 3-day epochs occurring on D2–D4, D7–D9 or D12–D14. Thick horizontal bars represent sample means. Individual circles represent all individual data points. Asterisks and NS denote statistically significant differences and lack thereof, respectively, for pairwise comparisons indicated by thin horizontal lines after correcting for multiple comparisons

Whereas CCol response gain remains stable under control conditions (Fig. 5b, gray line), we observed a rapid, bi-phasic change following cochlear synaptopathy, such that CCol response gain was depressed hours following noise exposure but then rose above baseline levels 1 day later, despite the substantial loss of auditory nerve input (Fig. 5b, red line, D2–D7). CCol response gain remained elevated for at least 14 days following cochlear synaptopathy, based on a subset of mice that underwent an extra week of daily imaging (Fig. 5c, $n = 5$). By compiling measurements of response gain in each mouse from each individual imaging session, we confirmed that CCol response gain was reduced by 83.3% hours after noise exposure (ANOVA main effect for group, pairwise comparison for control vs D1, $p = 0.00001$ after correcting for multiple comparisons, Fig. 5d). CCol response gain was significantly elevated above control levels at D2–D4, D7–D9 and D12–D14 time points (78.2%, 38.8% and 60.5%, respectively, $p < 0.05$ for each pairwise comparison after correcting for multiple comparisons). We found that the CCol gain elevation remained stable over time, as no significant differences were noted in D2–D4 to D12–D14 or D7–D9 to D12–D14 contrasts ($-9.9\%$ and $+15.6\%$, respectively, pairwise comparisons, $p > 0.39$ for each after correcting for multiple comparisons).

## Discussion

Central auditory neurons compensate for a sudden loss of input from the ear by increasing intrinsic excitability[43,44] and modifying both the sub-unit composition and functional response properties of excitatory and inhibitory synapses[45–52]. Collectively, these changes function as a central amplifier that increases the neural gain on diminished afferent inputs from the auditory periphery. At the level of single units or population responses in intact preparations, increased central gain can manifest as elevated spontaneous firing rates, increased spike synchronization, disinhibition and steeper rising slopes in sound level growth functions[3,53]. At the level of auditory perception, increased central gain may provide the means to maintain relatively normal perceptual thresholds and basic sound awareness even with an extreme (>90%) loss of peripheral afferents that renders the ABR grossly abnormal or absent altogether[41,54–56].

Increased central gain in the auditory system is sometimes described as a form of homeostatic plasticity, though it remains to be seen how well this label fits. Homeostatic plasticity is a negative feedback process that stabilizes neural activity levels following input perturbations. Homeostatic mechanisms modify excitatory and inhibitory synapses over a period of hours or days to offset input perturbations and gradually restore spiking activity back to baseline levels[57]. Central changes in auditory gain also offset a loss of input, but have an uncertain connection to homeostatic plasticity because studies have largely been based on acute measurements from unspecified cell types in separate deprived and control groups without ex vivo analysis of the underlying synaptic changes (but see refs. [48,49,58]). Recent work in the sensory-deprived somatosensory and visual cortex have identified shifting contributions from Hebbian and homeostatic plasticity mechanisms that drive increased excitability over the time course of deprivation[59], even between neighboring cell types[60], most notably in this context between the different types of L5 cortical projection neurons[61]. Understanding the mechanisms underlying increased central gain would benefit from the application of chronic 2-photon imaging from identified cell types followed by ex vivo recordings to determine whether underlying synaptic changes reflect homeostatic signaling pathways, Hebbian plasticity pathways or something else entirely[62].

By monitoring day-to-day changes in the activity of an anatomically defined cortical output neuron before and after sensory deprivation, the data described here provide new insights into the dynamics of compensatory plasticity (despite revealing little about underlying mechanisms). CCol growth functions rebounded above baseline levels and remained elevated through the duration of the experiment and therefore the changes described here were not strictly consistent with a homeostatic plasticity process. Although it is possible that CCol gain enhancement would eventually return to baseline, we recently reported that intracortical inhibition from parvalbumin (PV)-expressing GABAergic interneurons remains significantly depressed relative to pre-exposure baseline levels for as long as 45 days following cochlear synaptopathy[58], suggesting that the increased response growth functions observed in L5 neurons could remain elevated even at longer recovery times. This study reported that PV-mediated intracortical inhibition was reduced by as much as 50% over the same 14-day period studied here, during which time we observed a Hebbian-like enhancement of responses to low-frequency tones that stimulate undamaged regions of the cochlea. Interestingly, recordings from these unidentified ACtx regular spiking units found that the gain in sound level growth functions were only elevated during the first 2 days after noise exposure, far shorter than the 2 weeks of increased gain observed here in CCol neurons.

In the visual cortex, destruction of vestibular inputs leads to a sustained potentiation of L5 outputs to subcortical oculomotor nuclei to enable adaptive behavioral modifications in the optokinetic reflex[37]. In the somatosensory cortex, removal of the preferred whisker input also induces disinhibition[59] and potentiation of non-deprived whisker inputs in intrinsic bursting L5 neurons, but not neighboring L5 regular spiking neurons[61]. In ACtx, L5 neurons that project to the IC are intrinsic bursting, whereas neighboring L5 neurons that, for example, project to the contralateral hemisphere are regular spiking[15,17,63]. If the effects of sensory deprivation on L5 neurons in the ACtx parallel descriptions in other sensory cortices, the Hebbian component of plasticity in L5 CCol neurons may be expressed to a higher degree than neighboring cell types, producing a sustained potentiation of responses to spared sensory inputs following deprivation, particularly when PV-mediated intracortical inhibition is reduced.

Enhanced central gain is a hallmark of central auditory changes following noise-induced hearing loss, and has been linked to hyper-synchronization, dysrhythmia and associated perceptual disorders including hyperacusis and tinnitus[53,64]. Tinnitus is more than just a perceptual disorder, as subjects often report increased anxiety, stress and other complex and heterogeneous forms of mood dysregulation[65,66]. Aberrant activity in human subjects with tinnitus or animal models of tinnitus is observed far beyond the central auditory pathway and has been specifically linked to abnormally strong coupling of an extended network of brain areas including the ACtx, inferior colliculus, striatum and amygdala[67,68]. As these are the very same brain areas innervated by the L5 projection neurons studied here, one clear implication is that the increased sensory gain in these far-ranging ACtx corticofugal output neurons could be a key contributor to driving hyperexcitability and strong functional coupling in a distributed brain network underlying tinnitus.

## Methods

**Animal subjects**. Adult CBA/CaJ mice of either sex were used for all experiments in the study. All procedures were approved by the Massachusetts Eye and Ear Infirmary Animal Care and Use Committee and followed the guidelines established by the National Institute of Health for the care and use of laboratory animals.

**Virus injections**. Mice 6–8 weeks of either sex were anesthetized using isoflurane in oxygen (5% induction; 1.5–2% maintenance), with core body temperature maintained at 36.5°. Virus solution was backfilled into a pulled glass capillary pipette and injected into the target brain area at 15 nl/min using an automated injection system (Stoelting). For CCol axon imaging, we opened two small burr holes in the skull (0.5–1 mm diameter each) along the caudal-rostral extent of the squamosal suture that overlies the ACtx. After inserting the pipette 0.5 mm into the cortex, we then injected 250 nl of undiluted AAV5.Syn.GCaMP6s.WPRE.SV40 (UPENN Vector Core). The virus incubated for approximately 3–4 weeks before imaging began. For axon tracing experiments, we injected 500 nl of undiluted CAV2-Cre 0.5 mm deep at three equally spaced sites along the medial-lateral extent of the IC (Universitat Autònoma de Barcelona Vector Core) in two C57BL/6J mice (aged 6–8 weeks). The following day, we injected a cre-dependent GCaMP virus into the ACtx using the same injection protocol listed above (AAV1.Flex.GCaMP6s) and allowed the virus to incubate for 4–6 weeks before sectioning the brain. Following injections, a dab of antibiotic ointment was applied to each burr hole and the craniotomies were sealed with an ultraviolet (UV)-curing cement (Flow-It ALC Flowable Composite). The wound was closed and mice were injected with an analgesic (Buprenex, 0.05 mg/kg and Meloxicam, 0.1 mg/kg) before recovering in a warmed chamber.

**Chronic imaging preparation**. Cranial windows were made by etching glass cover slips in piranha solution ($H_2O_2$ mixed with $H_2SO_4$ in a 3:1 ratio) and stored in 70% ethanol. A 4 mm diameter cover slip was centered and affixed to a 3 mm cover slip (#1 thickness, Warner Instruments) using a transparent, UV-cured adhesive (Norland Products). Windows were stored in double deionized water and rinsed with sterile saline before use.

For Cranial window implantation surgery, animals were anesthetized with isoflurane in oxygen (5% induction; 1.5–2% maintenance). Dexamethasone sodium phosphate was administered to reduce brain edema (2 mg/kg, intramuscular). After removing the periosteum from the dorsal surface of the skull, an etchant (C&B Metabond) was applied for 30 s to create a better adhesive surface. A custom titanium headplate (iMaterialise) was bonded to the dorsal surface of the skull with dental cement (C&B Metabond). In accordance with a published protocol on chronic cranial window surgical procedure[38], we made a 3 mm circular craniotomy atop the IC with a pneumatic dental drill and diamond burr (head diameter 1/10 mm, NeoDiamond–Microcopy Dental). Once liberated, the bone flap was removed with great care and continuously irrigated with saline to avoid rupturing the pial vessels underneath. The cranial window was then lowered into place using a three-dimensional manipulator and bonded to the surrounding regions of the skull to create a hermetic seal. Post-operative injections of Buprenex (0.05 mg/kg) and Meloxicam (0.1 mg/kg) were administered and the mice were allowed to recover in a warmed chamber. Imaging began 5–7 days following recovery from surgery.

**Widefield calcium imaging**. Calcium imaging was performed in awake, head-fixed mice inside of a light- and sound-attenuating chamber mounted to an isolated optical table (Thorlabs). Blue light illumination was supplied in epifluorescence configuration from a broadband arc lamp (Lumen Dynamics) passed through a filter cube housing an excitation filter (482 ± 9 nm), dichroic mirror (reflection band: 350–488 nm; transmission band: 502–950 nm) and emission filter (520 ± 14 nm, Thorlabs) and focused on the surface of the IC with a 4×/ 0.13 NA objective (Nikon). Images (1392 × 1040 pixels) were acquired with a 1.4 Megapixel CCD camera and transferred to a PC via a Gigabit Ethernet interface to a framegrabber PCI card (Thorlabs). Image acquisition was hardware-triggered at 10 frames/s using a TTL pulse train synched to stimulus generation.

**Stimulus presentation**. Stimuli were generated with a 24-bit digital-to-analog converter (National Instruments model PXI 4461) and presented via a free-field tweeter (Vifa) positioned 10 cm from the left (contralateral) ear canal. Stimuli were calibrated before recording with a wideband ultrasonic acoustic sensor (Knowles Acoustics, model SPM0204UD5). Broadband noise bursts (50 ms duration, 4 ms raised cosine onset/offset ramps) were pseudorandomly presented between 20 and 80 dB SPL in 10 dB increments (50 repetitions per stimulus). Trial duration was 2 s.

**Imaging data analysis**. Images were first downsampled by a factor of 4 using bicubic interpolation. A region of interest (ROI) was positioned over an IC region with maximum CCol fluorescence that did not include surface blood vessels. Exact ROI dimensions varied between mice depending on blood vessel patterns and craniotomy location (100 × 100 ± 50 pixels) but was fixed in size and position across imaging sessions for a given animal. Population GCaMP responses were computed from the mean of all pixels within the ROI. Mice with low GCaMP expression in the cranial window during the initial baseline period were excluded from all analyses.

After averaging across trials, we computed the pre-stimulus fluorescence level ($F_0$) as the mean fluorescence across a 0.5 s period immediately prior to stimulus onset. We then expressed the fractional change in fluorescence (($F - F_0$)/$F_0$) for each frame ($F$). For each sound level, response amplitude was defined as the peak of the fractional change response, expressed as a percent change from baseline. A linear model was used to regress the response amplitudes on each day to the mean

response amplitude from the first two baseline imaging sessions. The regression was limited to the region of linear growth (40–80 dB SPL) to improve the goodness of fit ($R^2$) across all conditions. The slope of this least-squares fit ($m$) was used to quantify the degree of divisive ($m < 1$) or multiplicative ($m > 1$) gain changes across imaging days.

**Acoustic over-exposure**. Mice were exposed to an octave band of noise (8–16 kHz) presented at 100 dB SPL for 2 h. During exposures, animals were awake and unrestrained within a $12 \times 16 \times 16$ cm, acoustically transparent cage. The cage was suspended directly below the horn of the sound-delivery loudspeaker in a reverberant chamber. Noise calibration to target SPL was performed immediately before each exposure session.

**Cochlear function tests**. Mice were anesthetized with ketamine and xylazine (100/10 mg/kg for ketamine/xylazine, respectively, with boosters of 50 mg/kg ketamine given as needed). Core body temperature was maintained at 36.5° with a homeothermic blanket system. Acoustic stimuli were presented via in-ear acoustic assemblies consisting of two miniature dynamic earphones (CUI CDMG15008–03A) and an electret condenser microphone (Knowles FG-23339-PO7) coupled to a probe tube. Stimuli were calibrated in the ear canal in each mouse before recording.

ABR stimuli were tone bursts (8, 11.3, 16, 22.6 and 32 kHz) or white noise bursts (0–50 kHz), 5 ms duration with a 0.5 ms rise-fall time delivered at 27 Hz, and alternated in polarity to the left ear. Intensity was incremented in 5 dB steps, from 20 to 80 dB SPL. ABRs were measured with subdermal needle electrodes positioned beneath both pinna (+ and −) and the base of the tail (ground). Responses were amplified (gain = 10,000), filtered (0.3–3 kHz) and averaged (1024 repeats per level). ABR threshold was defined as the lowest stimulus level at which a repeatable wave 1 could be identified.

DPOAEs were measured in the ear canal using primary tones with a frequency ratio of 1.2, with the level of the $f_2$ primary set to be 10 dB less than $f_1$ level, incremented together in 5 dB steps. The $2f_1$–$f_2$ DPOAE amplitude and surrounding noise floor were extracted. DPOAE threshold was defined as the lowest of at least two consecutive $f_2$ levels for which the DPOAE amplitude was at least 2 standard deviations greater than the noise floor. All treated animals underwent rounds of DPOAE and ABR testing with tones before, 2 days and approximately 14 days after noise exposure. ABR to white noise bursts were measured every other day beginning ether 2 days before noise exposure ($n = 3$) or the day before noise exposure ($n = 3$), for a total of 4–5 ABR test sessions for a given mouse.

**Visualization of corticofugal axons**. Deeply anesthetized mice were perfused transcardially with 0.01 M phosphate-buffered saline (PBS; pH = 7.4) followed by 4% paraformaldehyde in 0.01 M PBS. Brains were removed and stored in 4% paraformadehyde for 12 h before transferring to cryoprotectant (30% sucrose in 0.01 M PBS) for at least 48 h. Sections (40 μm thick) were cut using a cryostat (Leica), mounted on glass slides and coverslipped using Vectashield Mounting Medium with DAPI (Vector Labs). ACtx cell bodies and distribution of CCol axons were visualized and photographed using an epifluorescence microscope (Leica).

**Cochlear histology and synapse quantification**. Cochleae were dissected and perfused through the round window and oval window with 4% paraformaldehyde in PBS, then post-fixed in the same solution. Cochleae were dissected into half-turns for whole-mount processing. Immunostaining began with a blocking buffer (PBS with 5% normal goat or donkey serum and 0.2–1% Triton X-100) for 1 to 3 h at room temperature and followed by incubation with a combination of the following primary antibodies: (1) rabbit anti-CtBP2 (BD Biosciences, Catalog No. 612044) at 1:100, (2) rabbit anti-myosin VIIa (Proteus Biosciences, Catalog No. 25–6790) at 1:200, (3) mouse anti-GluR2 (Millipore, Catalog No. AB1768-I) at 1:2000. Lengths of cochlear whole mounts were measured and converted to cochlear frequency. Confocal $z$-stacks from each ear were obtained in the inner hair cell area using a high-resolution glycerin-immersion objective (63×) and ×3.18 digital zoom with a 0.25 μm $z$-spacing on a Leica SP5 confocal microscope. For each stack, the $z$-planes imaged included all synaptic elements in the $x$–$y$ field of view. Image stacks were imported to image-processing software (Amira, Visage Imaging), where synaptic ribbons, glutamate receptor patches, and inner hair cells were counted.

**Statistical analyses**. Statistical analyses were performed in Matlab (Mathworks). Descriptive statistics are provided as mean ± SEM. Inferential statistics between control and noise-exposed samples were performed with two-tailed tests of unmatched samples (between-subject ANOVA or unpaired $t$-tests). Statistical contrasts over the noise exposure period were performed with a repeated measures ANOVA. All post-hoc pairwise comparisons were corrected with Bonferroni–Holm to account for type-I error inflation due to multiple comparisons.

**Data availability**. All data and software scripts that support the findings of this study can be made available from the corresponding author upon reasonable request.

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

## Acknowledgements

We thank the UAB Vector Core, GENIE Program, the Janelia Farm Research Campus and Penn Vector Core in the Gene Therapy Program of the University of Pennsylvania for providing the GCaMP6s and CAV2-Cre reagents for this project. We thank A.E. Hight for designing the head fixation hardware. We thank J. Dahmen, K. Kuchibhotla and C. Harvey for guidance on the surgical preparation for chronic imaging. We thank S. Kujawa and A. Parthasarathy for their contributions to cochlear synapse quantification. This work was supported by NIDCD R01 DC009836 (to D.B.P), NIDCD F32 DC015376 (to R.S.W.) and a Herchel Smith Graduate fellowship (to M.M.A.)

## Author contributions

M.M.A., R.S.W. and D.B.P. designed all experiments. M.M.A. and R.S.W. performed the experiments. M.M.A. collected and analyzed all data. K.E.H. wrote custom scripts for data acquisition. M.M.A. and D.B.P. wrote the manuscript.

## Additional information

**Competing interests:** The authors declare no competing interests.

