## [Peer Review File · Nature Communications]

Reviewers' comments:

Reviewer #1 (Remarks to the Author):

In Asokan et al 2017, the authors present two major advancements in their investigation of plasticity of corticocollicular (CCol) auditory neurons. The first is the development of a new approach to probe long-term changes in activity levels of CCol neurons in response to sound trauma, in isolation from other cortical neuron subtypes. The authors use a non-selective viral vector to express GCaMP6 in neurons of Layer V (mostly, also layer I-III) but then target the CCol neurons for chronic activity imaging in their axons where they terminate in the dorsal and external cortex of the inferior colliculus. This approach is technically challenging, but has the benefits of allowing chronic imaging through an implanted window, imaging of surface axons for improved resolution, and targeting of the CCol neurons but not other neurons such as callosal neurons. The second advancement is the finding that auditory CCol neurons demonstrate a pattern of plasticity in response to noise trauma that is quite different from the patterns simultaneously occurring in the ascending auditory system, and therefore due to other forms of 'homeostatic' plasticity. The authors do not describe a mechanism for this plasticity.

The authors use a degree of noise trauma that has been receiving much attention and study recently that induces noise damage termed 'hidden hearing loss', because after an initial reduction in auditory system function as assessed by ABRs, the system appears to return to normal. However, a significant loss of afferent synapses actually occurs very rapidly in the cochlea, initiating major compensatory neuronal activity changes in the brain that are just beginning to be understood. This work is very relevant for humans, as the noise trauma occurs at common levels of sound in the workplace or for recreation. The authors first replicate this type of noise damage, then show that the efferent CCol neurons follow a very different pattern of plasticity compared to ascending auditory neurons, implicating a CCol-specific mechanism of plasticity.

This work is well written and clear, and figures are well presented. The data is technically sound. The statistics detailed are appropriate, except for the additional detail required for figure 3 (see further comments below). The results will be most important for neuroscientists with an interest in central plasticity especially in descending systems, and for auditory neuroscientists with an interest in noise-induced hearing loss.

Details:

1. The viral vector injected into the cortex is non-selective. The authors describe this as a strength of their study, so that the entire population of CCol neurons can be studied simultaneously. The CCol neurons can then be studied apart from other L5 neurons such as cortico-cortico collosal neurons by targeting the axons as they project to the IC (or other region, this happens to be an accessible area for imaging). However, it is not clear whether the neurons being investigated are themselves a subpopulation of CCol neurons, or if the same neurons have axonal branches to other subcortical brain regions including those shown in figure 1B. Describe what is known about this. If the same neurons project throughout the auditory system (and may therefore have the same plastic changes while projecting to many other neuronal targets) it gives the results more weight. Otherwise,

effects may be limited to the external IC and less broadly interesting.

2. This paper places much weight on the rapidly changing patterns of plasticity shown in CCol neurons that fluctuate over a few days, as a contrast to the well-described synapse loss and plasticity in the ascending auditory system. The authors should demonstrate that they induce the same patterns of plasticity using this noise damage protocol in the ascending auditory system at a neuronal level, in order to draw conclusions regarding the different patterns seen in their current data. This is especially true because the imaging is done on a shorter and more fine-grained time scale (daily from -2 to +6 days following noise damage) compared to auditory function behavioral testing (pre-exposure, +1 day, and 2 weeks) that is used to prove that they are using a 'hidden hearing loss' noise damage protocol. The characteristic patterns of neuronal changes in the ascending system could be shown in a few ways:

a. Demonstrate cochlear synapse loss (a critical feature of hidden hearing loss) beginning at ~1 day after exposure, as shown in Kujawa and Liberman 2009. An immunolabel with synaptic markers is fine.

b. OR: demonstrate the expected patterns of growth of sound-evoked activity in the ascending auditory system (especially cortex) at 1 through 6 days following exposure, to demonstrate that while other neurons of the auditory system including the auditory cortex have a greatly increased gain that is long lasting, the CCol gain returns to baseline by about 5 days after noise trauma.

3. In the results text for figure 2, the authors detail the three phases of hearing activity for which they were able to pair CCol axon imaging. The match between stages i) and ii) with the imaging is fine, but the authors do not prove that they image during iii), a period with normal cochlear thresholds but a permanent loss of cochlear afferent synapses (same critique as above, the synaptic loss is not proven). This could be addressed with histology (eg GluA2/3 R staining) to show synapse loss to demonstrate that the ABR wave 1 reduction at 1 day post noise-induction was not caused by some other mechanism.

4. Figure 3 requires significant clarification of methods and added statistical detail, especially because the differences between panels with and without significant changes are very difficult to make out by eye. Please clarify the stats used to designate 'potentiated' vs 'non-potentiated' in figure 3A (and what 'greatly reduced' requires, line 180), and whether deviation from monotonic is a criterion. Then, for figure 3B, detail the slopes measured, and how far away from 1 the slope must be in order to have either multiplicative or divisive scaling (eg for 'strong divisive gain', line 194).

5. The panels in figure 4A showing the response gain of individual mice all show the same pattern of a reduction in gain immediately following noise exposure (day 0), then an increase in gain, then a return to near baseline. However, the response does not plateau, and plots seem to be continuing to change in either direction. This suggests some sort of oscillation of the response gain over time. Adding a few more data points would make the current results more convincing (to the point of plateau at baseline), but 1) would require a total repeat of a technically challenging experiment and 2) wouldn't significantly change the conclusions, so additional experiments are not required for publication. Any follow up work

should extend this timeline if technically possible.

6. Line 179 (results section for figure 3A) typo – delete 'hours'

Reviewer #2 (Remarks to the Author):

While the plasticity of the ascending central auditory pathway has been intensively investigated, little is known how about plasticity in the descending pathways. In this study the authors report that the descending projections from deep cortical layers to the inferior colliculus becomes more sensitive to sound during 1-3 days after cochlear noise damage. This increased sensitivity is temporary as it only lasts a few days after which the corticocollicular responses decrease again.

This is a very focused study on 5 animals using in vivo calcium imaging of axons in the superficial layers of the IC to monitor sound-evoked activity of the corticocollicular system. The approach is novel and elegant. The main result that the corticocollicular system dynamically adjusts its 'gain' is novel and important.

My main critiques are that the link between the time course of temporary and permanent hearing loss and the changes observed in corticocollicular activity is weak. Without this link, the results loose power. I also have issues with how the data are analyzed to support the major claim of a dynamic gain adjustment.

The effects of noise exposure on DPOAE and ABRs are shown before noise exposure and 1 day and 2 week afterwards (Fig 2). Since the imaging of CCol activity covers the 6 days after noise exposure, DPOAE and ABR measurements should be shown for these 6 days, not at 2 weeks. Without ABRs from day 2, 3, 4, 5, and 6 the time course of CCol activity changes are difficult to interpret.

Because of different stimuli it is difficult to relate the results in Fig. 2, and Fig.3,4. For ABRs tone pips are used to show the frequency dependence of the hearing loss while for calcium responses of CCol fibers white noise is used. ABR responses to white noise are more relevant for interpreting the time course of CCol response changes and they need to be shown.

Except for day 0, the slopes in Figure 3 are strongly dominated by the single measurement at 80 dB. I assume none of the values at 20-70 dB is significantly different compared to day -2. In any case, an appropriate statistical analysis of the 80 dB measurements at day-2 and subsequent days is necessary. If they are not significantly different from Day -2 (perhaps on day 4), then the increase or decrease in gain is also not significant, even if the gain (slope) change seems to appear large. None of the values at 50-70 dB intensities at days 1-6 seems to be statistically different from day -2, even some changes would be expected based on figure 2 E. Based on this, we do not know what points in Fig 4A significantly deviate from a gain of 1 (day -2). It follows that the basis for the statistical analysis in Fig. 4B is

questionable and thus that the main conclusion of the paper is not well supported.

The authors state repeatedly in the abstract and throughout the paper, that CCol responses stabilized at baseline levels on day 6. However, 2 of the 5 animals show an upward trend at day 6 (mouse 1 and 5). More importantly, no data are presented that show any stabilization of responses after day 6. For example, the downward trend in at later days in Mouse 2,3 and 4 may continue or the upward trend in Mouse 1 and 5 may continue. Without showing stable responses over several days we simply do not know whether the responses are stable or not.

It is not clear whether the results shown in Fig. 2 are from a single animal or from a population

Which mouse is the one shown in Fig. 3? It seems Mouse 3, but should be specified

The effect of CCol input gain on the IC will depend on the potential changes in synaptic efficacy and excitability of the IC neurons postsynaptic to the CCol afferents. This could be addresses with chronic recordings of local field potentials.

The maximum response amplitude in Figure 1G is less than 1.5% but in figure 3 it is more than 6%. What animal was shown in fig 1? Is there a relationship between response amplitudes and hearing loss induced changes in CCol activity?

Discussion line 233-253. It is not clear how this paragraph helps to understand and interpret the current study. It reads like a review without a link to the present study. Almost the same can be said for the next paragraph (line 254 to 274)

Line 152: include frequency range and intensity of the octave-wide noise

Line 218: "... precise homeostatic adjustment described here in CCol population responses..." why was this adjustment 'precise'?

line 371: The imaging approach in the present study is not "transcranial"

line 436: 40mm → 40um

line 497, 510: reference 7 and 12 are the same

line 532: ref 22 does not have journal name

Does ref 23 really show evidence for changes in the level or subunit composition of neurotransmitter receptors?

Reviewer #3 (Remarks to the Author):

Asokan et al., presents a series of data that is examining Ccol axon responses following auditory insult. While the data presented in this study are potentially interesting, there are a few technical clarifications that are necessary before publication. Additionally, there are

several places in the manuscript where the wording is imprecise and thus it seems as if the results are overstated. Importantly, I think that this study would greatly benefit from the inclusion of sham-exposed controls to understand some of the variability observed in their data and separate variability due to experimental preparation and auditory insult.

Specific Comments

1. GCaMP was originally described in Tian et al., 2009 to cause abnormal activity once it enters the nucleus of the cell body. While this is less of a problem with GCaMP6, it still does occur and standard practice is to exclude cells with nuclear filling. How do the authors know that the axons they are recording GCaMP signals from are in healthy cells? This is particularly important given the wide range of responses that the authors see across animals in figure 4, which could be influenced by varying levels of GCaMP expression.
2. What are the expected levels of variability from GCaMP axon imaging with widefield? Is there at least a study that the authors can refer to using this technique to give the reader an idea of how much variability is expected from the experimental preparation used?
3. In Xu et al., Nat. Neurosci, 2007 and Holtmaat et al., Nat. Protocols, 2009, it is shown that there is a strong upregulation of astrocytes and microglia in the first few weeks (2-4 weeks) after surgery. The authors here only wait 5 days between surgery and imaging. Could the authors please confirm that there is not a strong immune response in their preparations, which could contribute to variability in activity levels between animals (perhaps through immunostaining of their surgical preparations after 5 days or by providing control animals which would help separate effects of experimental preparation from effects due to auditory insult).
4. I do not see that authors have checked whether they actually create a loss of afferent synapses with their protocol. While the histology showing this would be nice, I think that their wording throughout should at least reflect that they have not checked. For example, in the abstract there is really no evidence in this paper that there is a compensation for a permanent loss of cochlear afferent synapses. Or in line 171-172, it is not clear that they are measuring during a period of permanent loss of cochlear afferent synapses.
5. In line 175-177, I think that the statement they make here should be quantified. When I look at figure 4, it is not clear to me that the pattern is repeatable, or that animals 3, 4, or 5 have a stabilization, which to me would mean being at baseline for multiple time points. If they want to say this, they need to quantify what they mean by each part of the statement and provide analyses with statistics to back up their statements. Given the strong variability in their signals, I think that a few control animals would greatly strengthen their ability to draw conclusions from their data (and provide a robust way to do statistics).
6. What do the authors mean by homeostatic plasticity? Just that activity returns? Typically, recovery of activity that is termed homeostatic is associated with a homeostatic mechanism (like synaptic scaling for example). I do not object to the use of the term without an accompanying homeostatic mechanism, but I just would like to know why do they think this is a homeostatic response instead of a Hebbian or adaptation response - especially give its time course of days? Could they please include this in the discussion?
7. The authors use the term 'multiplicative scaling' in line 187 and again in the figure legend. What exactly do they mean by that? Given that they are using the term 'homeostatic plasticity', I assume this term to mean that there is a shift of a given distribution by a multiplicative factor (typically miniature EPSCs or IPSCs), such that you

take the distribution before deprivation, multiply it by a factor and then it overlays with the after deprivation distribution. It is fine if they want to apply this term to their response curves in figure 3, but then the entire distribution should shift. All that is changing in both figure 3A and B is the 80dB data point, which to me, does not seem like multiplicative scaling. If they want to use this term (although I think it is a pretty loaded term so personally I would use different wording), it would be helpful if they defined it and quantified how they decided their changes were multiplicative.

8. There is a huge range in the individual mouse responses in figure 4. Is there any correlation with the behavioral measures (DPOAE and ABR) and the GCaMP response of the animal? If not, could they mention in the discussion why they think there is not a correlation between activity and behavior.

9. Are there significant differences in figure 2d? could they please label them in the figure with asterisks or NS? Please label all statistics in the figure legends and use standard marking on the figures.

10. In line 218, Could the authors please elaborate about what they consider 'precise' about the homeostatic adjustment? Is it the timing or the degree (or something else)?

Additionally, the authors contrast the current study to central gain potentiation, which "over-shoots the mark" (line 220) elevating central excitability. The authors data also overshoot the mark and it is not clear the mechanism by which that happens, but there is no obvious reason to me to assume that it is not through a change in excitability. Could they please clarify?

11. In the section of the discussion on Central Gain Enhancement (lines 233-252), it is not clear to me how this section relates to the current data, which is not mentioned in the entirety of this section. I would either explain how this is relevant to the current study, or remove it. Same is true for the next section on Hierarchical Regulation (lines 255-274).

12. Again in line 298, only mouse 2 (and maybe 1) really stabilizes at baseline levels (assuming stabilization is multiple time points). So I would remove this emphasis or provide statistics of which mice and timepoints are not statistically significantly different from baseline.

13. In lines 298-302, the authors suggest that their time course fits the signature of a homeostatic process that dips in response, compensates at 24 hours, then overshoots before coming back to baseline. In Hengen et al., the paper the authors cite as the basis for this signature, Hengen et al. specifically state that there is not an overshoot in their data:

"Crucially, over the next 2 days of MD (MD3–MD4), firing rates rebounded and by MD5–MD6 were indistinguishable from baseline. Although mean firing rates were ~9% higher on MD6 (P32) relative to baseline (P26), this increase was within the range of variation in the control hemisphere (Figure 2C) and was not significant ($p = 0.98$, Figure 2D, Tukey-Kramer test)." (Hengen et al., 2013)

This section should be reworded to either include references that support their description or to reflect what is reported in Hengen et al., 2013.

14. From the methods, it seems that the authors see the strongest effect 30-60 minutes after their noise exposure. Given that they do not do any behavior measures until at least one day later, how do they know that the animals are actually suffering from hearing loss so quickly? Could they not just have a short-term adaptation, which goes away? What would

happen if they record 6 or 12 hours after exposure or if they measured the ABR 30 minutes after the noise exposure? If the animals are not suffering from hearing damage, is the response actually homeostatic?

15. Related to that last point, can the authors rule out the effects of stress as a result of loud noise exposure? For example, are there changes in response to the cerebellum activity levels (as in figure 1) immediately after noise exposure?

16. When exactly were the animals injected with ketamine/xylazine and does that affect their GCaMP activity levels?

17. In the methods, it says that the ABR was done 2 days after noise exposure and in the results and figures, 1 day after exposure. Could the authors clarify this?

18. What frequency did the authors use in figure 3A-B?

Reviewer #1

This work is well written and clear, and figures are well presented. The data is technically sound. The statistics detailed are appropriate, except for the additional detail required for figure 3 (see further comments below). The results will be most important for neuroscientists with an interest in central plasticity especially in descending systems, and for auditory neuroscientists with an interest in noise-induced hearing loss.

>>> We thank the reviewers for their thoughtful comments and critiques. Since receiving the reviews in July, we have worked hard to perform many new experiments suggested by the reviewers. All told, we have more than quadrupled the sample size from our original submission. We thank the reviewers for suggesting that we dig deeper and perform the additional experiments as they have undoubtedly improved the impact and accuracy of our work. Major new additions include:

- New intersectional anatomy experiments to demonstrate that CCol neurons are members of a broad class of Layer 5b projection system with widespread projections in the forebrain and midbrain (Fig. 1a, as suggested by reviewer 1)
- New quantitative immunolabeling of primary cochlear afferent synapses 1 day and 14 days after moderate noise exposure to confirm the classic “synaptopathy” description (Fig. 3e-g, as suggested by reviewers 1 and 3).
- New measurements of daily changes in ABR thresholds and amplitudes using broadband stimuli following noise exposure to provide a more direct comparison with daily changes in corticofugal response gain (Fig. 4c and 5a-b, as suggested by reviewers 1,2).
- New measurements of the extended time course of CCol changes to include daily imaging for 14 days following noise exposure (Fig. 5c-d, as suggested by all reviewers)
- New measurements from a sham-exposed control group (Fig. 2 and Fig 5b-d, as suggested by reviewer 3).

1. The viral vector injected into the cortex is non-selective. The authors describe this as a strength of their study, so that the entire population of CCol neurons can be studied simultaneously. The CCol neurons can then be studied apart from other L5 neurons such as cortico-cortico callosal neurons by targeting the axons as they project to the IC (or other region, this happens to be an accessible area for imaging). However, it is not clear whether the neurons being investigated are themselves a subpopulation of CCol neurons, or if the same neurons have axonal branches to other subcortical brain regions including those shown in figure 1B. Describe what is known about this. If the same neurons project throughout the auditory system (and may therefore have the same plastic changes while projecting to many other neuronal targets) it gives the results more weight. Otherwise, effects may be limited to the external IC and less broadly interesting.

>>> As the reviewer points out, previous work has shown that ACTx projection neurons target a wide range of brain areas in the ipsilateral telencephalon, contralateral neocortex, ipsilateral thalamus, ipsilateral midbrain and even the earliest stages of sound processing in the ipsilateral cochlear nucleus. Is each downstream brain area innervated by a separate “type” of corticofugal projection neuron? Or does a single type of corticofugal neuron (e.g., a L5 corticofugal projection neuron) innervate multiple downstream areas? This has been addressed in a few anatomical studies by injecting retrograde tracers into two brain areas and quantifying the fraction of double-labeled neurons. However, this approach is plagued by challenges in “interpreting the negative”. If double-labeled neurons are not found, it could mean that neurons project to Area A but not Area B, or instead it might simply result from missing the innervation zone with one of the tracer injections. This is a major problem as it is impossible to fill the entire downstream brain structure with retrograde tracer without also impinging on nearby brain areas.

Intersectional virus strategies provide an elegant workaround. With this approach, a virus that solely infects axon terminals (e.g., Retro-AAV, CAV, or Rabies virus) can be injected into the inferior colliculus to express Cre-recombinase in all neurons that project to that area. Then, a Cre-dependent virus is injected into the ACTx to express a fluorescent protein throughout the entire axon field of these corticocollicular neurons. Of course axon terminals will be found in the IC itself, but the critical question is whether labeled axons from ACTx -> IC neurons are also found in other areas that were not targeted by the injection. If axon terminals are found in other brain areas, it provides strong evidence that at least some ACTx neurons that project to the IC must also project to other, potentially multiple downstream targets.

We performed this intersectional virus experiment in two mice. We injected CAV-Cre into the right inferior colliculus and a FLEX virus to express a fluorescent marker into the right ACTx (FLEX-GCaMP with a longer incubation time worked fine for this purpose). We observed that corticocollicular neurons also drop extensive networks of axon terminals in the ipsilateral medial geniculate body, ipsilateral striatum and ipsilateral lateral amygdala (Fig. 1a). Only very sparse axon labeling was observed in the corpus callosum or dorsal cochlear nucleus (though again, there are caveats in interpreting the negative, so these sections are not shown). The observation of dense axon terminal fields in other brain areas suggesting that at least some CCol neurons could also be described as corticothalamic, corticostriatal and corticoamygdalar neurons. As the reviewer notes, these findings increase the interest level in our work because it suggests that our approach is using the IC as a convenient location to quantify the dynamics of a wide-ranging class of projection neuron. Further to this, increased gain of excitatory inputs into limbic areas like the lateral amygdala can have interesting implications for common reports of stress, anxiety and mood dysregulation that accompany tinnitus. These new data are provided in Figure 1.

2. This paper places much weight on the rapidly changing patterns of plasticity shown in CCol neurons that fluctuate over a few days, as a contrast to the well-described synapse loss and plasticity in the ascending auditory system. The authors should demonstrate that they induce the same patterns of plasticity using this noise damage protocol in the ascending auditory system at a neuronal level, in order to draw conclusions regarding the different patterns seen in their current data. This is especially true because the imaging is done on a shorter and more fine-grained time scale (daily from -2 to +6 days following noise damage) compared to auditory function behavioral testing (pre-exposure, +1 day, and 2 weeks) that is used to prove that they are using a 'hidden hearing loss' noise damage protocol. The characteristic patterns of neuronal changes in the ascending system could be shown in a few ways:

- a. Demonstrate cochlear synapse loss (a critical feature of hidden hearing loss) beginning at ~1 day after exposure, as shown in Kujawa and Liberman 2009. An immunolabel with synaptic markers is fine.
- b. OR: demonstrate the expected patterns of growth of sound-evoked activity in the ascending auditory system (especially cortex) at 1 through 6 days following exposure, to demonstrate that while other neurons of the auditory system including the auditory cortex have a greatly increased gain that is long lasting, the CCol gain returns to baseline by about 5 days after noise trauma.

>>> Per option A, we have performed quantitative immunolabeling of primary afferent synapses onto inner hair cells in the high frequency cochlear base at 1 day and 14 days following this moderate noise exposure paradigm. In agreement with prior reports, we find that approximately 50% of the synapses are lost within 24 hours following noise exposure and this loss is stable across a later measurement time. These new data are provided in Figure 3e-g. In light of our revised analysis of the expanded corticofugal plasticity data set, we now are able to report that the response gain remains elevated up to 2 weeks after noise exposure. This agrees with previous findings in other cell types and alleviates the need to perform recordings from other cell types in the ascending pathway, per Option B.

3. In the results text for figure 2, the authors detail the three phases of hearing activity for which they were able to pair CCol axon imaging. The match between stages i) and ii) with the imaging is fine, but the authors do not prove that they image during iii), a period with normal cochlear thresholds but a permanent loss of cochlear afferent synapses (same critique as above, the synaptic loss is not proven). This could be addressed with histology (eg GluA2/3 R staining) to show synapse loss to demonstrate that the ABR wave 1 reduction at 1 day post noise-induction was not caused by some other mechanism.

>>> The findings from our new experiments no longer motivates the need to break the plasticity into three epochs. Rather, we see the strong immediate drop in response gain and the overshoot on the following days, but we don't see evidence for a return to baseline. Regardless, in a separate cohort of mice we measured ABR thresholds and ABR wave 1 amplitudes using the same broadband noise stimulus that was used to measure corticofugal plasticity (Fig. 4b). These new ABR measurements provided single day resolution of ABR changes during the same period of rapid corticofugal plasticity. In light of these new experiments, we can track changes in ABR threshold (Fig. 5a), ABR wave 1 (Fig. 5b) and corticofugal response gain (Fig 5b) with the same stimulus having now also shown that that 50% of the primary afferent fibers have already been eliminated in the cochlear base within 24 hours of noise exposure (Fig. 3g).

4. Figure 3 requires significant clarification of methods and added statistical detail, especially because the differences

between panels with and without significant changes are very difficult to make out by eye. Please clarify the stats used to designate 'potentiated' vs 'non-potentiated' in figure 3A (and what 'greatly reduced' requires, line 180), and whether deviation from monotonic is a criterion. Then, for figure 3B, detail the slopes measured, and how far away from 1 the slope must be in order to have either multiplicative or divisive scaling (eg for 'strong divisive gain', line 194).

>>> The example case shown in Figure 3 (now Fig. 4) has been changed. We now provide the 95% confidence intervals around the linear fit. Further, we have removed unnecessary adjectives from the text descriptions "greatly", "strong" etc. in favor of straightforward descriptions.

5. The panels in figure 4A showing the response gain of individual mice all show the same pattern of a reduction in gain immediately following noise exposure (day 0), then an increase in gain, then a return to near baseline. However, the response does not plateau, and plots seem to be continuing to change in either direction. This suggests some sort of oscillation of the response gain over time. Adding a few more data points would make the current results more convincing (to the point of plateau at baseline), but 1) would require a total repeat of a technically challenging experiment and 2) wouldn't significantly change the conclusions, so additional experiments are not required for publication. Any follow up work should extend this timeline if technically possible.

>>> We decided to bite the bullet and perform the same chronic imaging experiment on five new mice, this time extending the imaging period by another week so that we could track daily changes over a longer period. With these new data, it is clear that response gain wasn't returning to baseline within the originally described six-day period and the elevated gain was stable at 2 weeks. We thank the Reviewer for encouraging us to look more closely into what we interpreted to be a "steady state". The new data are shown in Fig. 5C and combined with the original data in Fig. 5B and 5D. Looking at the fourth set of data in Fig. 5D, it is clear that the few imaging sessions where the gain returned to baseline were not representative of a larger sample.

6. Line 179 (results section for figure 3A) typo – delete 'hours'

>>> Done

Reviewer #2

While the plasticity of the ascending central auditory pathway has been intensively investigated, little is known how about plasticity in the descending pathways. In this study the authors report that the descending projections from deep cortical layers to the inferior colliculus becomes more sensitive to sound during 1-3 days after cochlear noise damage. This increased sensitivity is temporary as it only lasts a few days after which the corticocollicular responses decrease again.

This is a very focused study on 5 animals using in vivo calcium imaging of axons in the superficial layers of the IC to monitor sound-evoked activity of the corticocollicular system. The approach is novel and elegant. The main result that the corticocollicular system dynamically adjusts its 'gain' is novel and important.

My main critiques are that the link between the time course of temporary and permanent hearing loss and the changes observed in corticocollicular activity is weak. Without this link, the results loose power. I also have issues with how the data are analyzed to support the major claim of a dynamic gain adjustment.

>>> We thank the reviewers for their thoughtful comments and critiques. Since receiving the reviews in July, we have worked hard to perform many new experiments suggested by the reviewers. All told, we have more than quadrupled the sample size from our original submission. We thank the reviewers for suggesting that we dig deeper and perform the additional experiments as they have undoubtedly improved the impact and accuracy of our work. Major new additions include:

- New intersectional anatomy experiments to demonstrate that CCol neurons are members of a broad class of Layer 5b projection system with widespread projections in the forebrain and midbrain (Fig. 1a, as suggested by reviewer 1)
- New quantitative immunolabeling of primary cochlear afferent synapses 1 day and 14 days after moderate noise exposure to confirm the classic "synaptopathy" description (Fig. 3e-g, as suggested by reviewers 1 and 3).
- New measurements of daily changes in ABR thresholds and amplitudes using broadband stimuli following noise exposure to provide a more direct comparison with daily changes in corticofugal response gain (Fig. 4c and 5a-b, as suggested by reviewers 1,2).
- New measurements of the extended time course of CCol changes to include daily imaging for 14 days following noise exposure (Fig. 5c-d, as suggested by all reviewers)
- New measurements from a sham-exposed control group (Fig. 2 and Fig 5b-d, as suggested by reviewer 3).

The effects of noise exposure on DPOAE and ABRs are shown before noise exposure and 1 day and 2 week afterwards (Fig 2). Since the imaging of CCol activity covers the 6 days after noise exposure, DPOAE and ABR measurements should be shown for these 6 days, not at 2 weeks. Without ABRs from day 2, 3, 4, 5, and 6 the time course of CCol activity changes are difficult to interpret.

>>> We now describe daily changes in ABR thresholds and wave 1 amplitude over the same six-day period described for the corticofugal imaging (Fig. 4b, 4c, 5a, 5b).

Because of different stimuli it is difficult to relate the results in Fig. 2, and Fig.3, 4. For ABRs tone pips are used to show the frequency dependence of the hearing loss while for calcium responses of CCol fibers white noise is used. ABR responses to white noise are more relevant for interpreting the time course of CCol response changes and they need to be shown.

>>> We performed these new ABR experiments using white noise bursts rather than tone bursts. As the reviewer notes, this facilitates a more direct comparison between peripheral and central measurements.

Except for day 0, the slopes in Figure 3 are strongly dominated by the single measurement at 80 dB. I assume none of the values at 20-70 dB is significantly different compared to day -2. In any case, an appropriate statistical analysis of the 80 dB measurements at day-2 and subsequent days is necessary. If they are not significantly different from Day -2 (perhaps on day 4), then the increase or decrease in gain is also not significant, even if the gain (slope) change seems to

appear large. None of the values at 50-70 dB intensities at days 1-6 seems to be statistically different from day -2, even some changes would be expected based on figure 2 E. Based on this, we do not know what points in Fig 4A significantly deviate from a gain of 1 (day -2). It follows that the basis for the statistical analysis in Fig. 4B is questionable and thus that the main conclusion of the paper is not well supported.

>>> In retrospect, this was a misleading example. To be fair, the response at 70 dB SPL was also above the line of unity, but we agree that the increased response was definitely greatest at 80 dB SPL. We chose this mouse because we wanted to include an example of a mouse where the gain returned to baseline. We now show a representative case in the newly added Fig. 2 to illustrate how the analysis was performed, which obviates the need to show example data. Instead, we now just present the mean CCol response growth functions across all animals (Figure 4D) and include the 95% confidence intervals on the regression plot (Fig. 4D, bottom) and the SEM across animals in the sound level growth functions (Fig. 4D, top). This seemed like the most straightforward way to communicate our findings.

With that said, we are not completely on board with the reviewer's interpretation that "the basis for the statistical analysis in Fig. 4B is questionable and thus that the main conclusion of the paper is not well supported". The main conclusion of the paper – both the last draft and the revision – is the response gain in this corticofugal projection neuron is paradoxically increased when the input from the auditory nerve is decreased. This is shown very clearly and quantifying response gain according to the slope of the linear fit is a more mathematically direct way to estimate gain than looking at significant differences for single sound levels because the slope term provides the multiplicative coefficient for scaling the response between the baseline period to any given day after sound exposure. In any system, gain describes the increase in output per unit increase in the input. We take the CCol response amplitude measured at baseline and again at various points after cochlear nerve damage. As the response gain cannot be estimated from a single input level, we fit the region of linear growth across the five highest sound levels for a given pair of imaging sessions with a 1st order polynomial and estimate the change in response gain according to the slope of the fit line. This is a standard approach used to quantify changes in cortical response gain after various types of manipulations (e.g., Olsen, Scanziani et al., Nature 2012; Nelson and Mooney, Neuron 2016; Guo, Polley et al., Neuron 2017). The goodness of fit for this linear estimate in our data is quite high ($R^2 = 0.8 \pm 0.03$, on average), so any changes in the slope are unlikely to arise solely from any one sound level.

On the other hand, yes, increasing the N from 5 to 10, increasing the imaging period from 7 to 14 days and improving the analysis as described above (SEM of growth functions, confidence intervals for single points, restricting the fit to the highest five sound levels) changed our conclusion that the gain increase completely reversed itself. In fact, the increased gain remains elevated from Day 2 to Day 14. This is easier to interpret, given persistent decreases in cortical inhibition following this same synaptopathy protocol (Resnik and Polley, eLife 2017) and is particularly interesting given that these projections innervate a wide range of downstream targets (Fig. 1a) that would themselves have to contend with an amped up excitatory input. For example, increased gain of these corticofugal excitatory inputs into limbic areas like the lateral amygdala can have interesting implications for common reports of stress, anxiety, mood dysregulation and misphonia following noise exposure. This is now included in the revised Discussion.

The authors state repeatedly in the abstract and throughout the paper, that CCol responses stabilized at baseline levels on day 6. However, 2 of the 5 animals show an upward trend at day 6 (mouse 1 and 5). More importantly, no data are presented that show any stabilization of responses after day 6. For example, the downward trend in at later days in Mouse 2,3 and 4 may continue or the upward trend in Mouse 1 and 5 may continue. Without showing stable responses over several days we simply do not know whether the responses are stable or not.

>>> Right, this was our motivation for performing all the new experiments. The reviewer was correct and we were wrong. Ultimately, we are glad that we got it right before publishing and we thank the Reviewer for pushing us to collect more data. As stated above, a persistently increased gain doesn't detract from the interest of the findings. To the contrary, it reveals a potentiated response in a far-ranging L5 output neuron that innervates many non-auditory brain areas. The revised (essentially, completely rewritten) Discussion compares these data to other ACTx cell types following this cochlear synaptopathy protocol and more broadly to the homeostatic plasticity literature, including parallel reports of sustained potentiation of L5 projection neurons in the somatosensory and visual cortex following peripheral sensory deprivation (pg. 12-13).

It is not clear whether the results shown in Fig. 2 are from a single animal or from a population

>>> The ABR and DPOAE data shown in Figure 3 (formerly Fig 2) are from a population. This has been clarified and the outcome of statistical tests are now indicated in the figure.

Which mouse is the one shown in Fig. 3? It seems Mouse 3, but should be specified

>>> Yes, that was Mouse 3. The individual line plots from the last version have been replaced with scatterplots in Figure 5D.

The effect of CCol input gain on the IC will depend on the potential changes in synaptic efficacy and excitability of the IC neurons postsynaptic to the CCol afferents. This could be addressed with chronic recordings of local field potentials.

>>> That's true, the net effect on the downstream neuron could depend on intrinsic changes in the IC neurons (e.g., changes in HCN channels), postsynaptic changes in IC excitatory receptors (e.g., AMPA receptor synaptic scaling), postsynaptic changes in IC inhibitory receptors (e.g., GABA receptor internalization) or additional changes in the presynaptic release properties from CCol axon terminals (e.g., vesicle packaging). Further, any of these changes could happen equivalently in excitatory and inhibitory IC neurons or instead could be differentially expressed wherein inhibitory interneurons express a different complement of changes than excitatory neurons to promote network plasticity phenomena like disinhibition. These synaptic changes could be accompanied by additional structural changes in neurite motility, extracellular matrix composition, even astrocytic regulation of neural excitability. All of these phenomena have been observed in various combinations in various systems. Clearly, too many things to test here and many would require measurement approaches beyond local field potentials to really nail down. The major advance in this study is that we developed a chronic imaging approach to track daily changes in its sensory-evoked excitability from a canonical cortical cell type following a well-controlled damage to afferent fibers in the peripheral nerve. There are many positive aspects of using chronic imaging approaches to study plasticity phenomena, but homing in on the underlying mechanisms often requires acute, ex vivo measurements. With all the new experiments that we have performed, we hope the reviewer will agree that the revised manuscript provides a complete description that nonetheless raises many questions for future investigation (which we address in the revised Discussion).

The maximum response amplitude in Figure 1G is less than 1.5% but in figure 3 it is more than 6%. What animal was shown in fig 1? Is there a relationship between response amplitudes and hearing loss induced changes in CCol activity?

>>> One of the sources behind the "wide range of responses" was purely analytic; in the original manuscript, we computed the DF/F for each trial and then averaged. In the revised manuscript, we average the trials before computing the DF/F. The latter approach is standard and has the effect of reducing measurement variability in the trial-averaged signal. To put a number on it, the coefficient of variation in the peak DF/F response amplitudes on the first day of imaging from the five original mice was 0.62 and the improved analysis method reduces the variance from the same trials in the same mice by nearly 50% (to 0.35). Looking across the data shown in the revised manuscript, the reviewer will appreciate that the maximum fractional change in the sound-evoked response is much more consistent across mice. Further to this, no, we did not observe any correlation between the baseline response amplitude and the mean gain increase measured on D2-4 across our mice ($n = 10$, Pearson $R = -0.23$; $p = 0.530$).

Discussion line 233-253. It is not clear how this paragraph helps to understand and interpret the current study. It reads like a review without a link to the present study. Almost the same can be said for the next paragraph (line 254 to 274)

>>> As the same point was raised by Reviewer 3, we removed these sections and instead focus the Discussion on the potential mechanisms that may underlie the changes we report and the potential perceptual consequences.

Line 152: include frequency range and intensity of the octave-wide noise

>>> Done

Line 218: "... precise homeostatic adjustment described here in CCol population responses..." why was this adjustment 'precise'?

>>> This has been removed in the revised manuscript.

line 371: The imaging approach in the present study is not “transcranial”

>>> Thanks, this has been corrected.

line 436: 40mm → 40um

>>> This has been corrected.

line 497, 510: reference 7 and 12 are the same

>>> This has been corrected.

line 532: ref 22 does not have journal name

>>> This has been corrected.

Does ref 23 really show evidence for changes in the level or subunit composition of neurotransmitter receptors?

>>> Reference 23 (Winkowski and Knudsen, Nature 2006) was not cited in this context. Perhaps the reviewer was referring to the former reference 13 (Middleton et al. PNAS 2011)? This study demonstrates decreased GABAergic inhibition but the reviewer was correct that it does not drill down to changes in the GABA receptor itself. We have removed this reference.

Reviewer #3

Asokan et al., presents a series of data that is examining Ccol axon responses following auditory insult. While the data presented in this study are potentially interesting, there are a few technical clarifications that are necessary before publication. Additionally, there are several places in the manuscript where the wording is imprecise and thus it seems as if the results are overstated. Importantly, I think that this study would greatly benefit from the inclusion of sham-exposed controls to understand some of the variability observed in their data and separate variability due to experimental preparation and auditory insult.

>>> We thank the reviewers for their thoughtful comments and critiques. Since receiving the reviews in July, we have worked hard to perform many new experiments suggested by the reviewers. All told, we have more than quadrupled the sample size from our original submission. We thank the reviewers for suggesting that we dig deeper and perform the additional experiments as they have undoubtedly improved the impact and accuracy of our work. Major new additions include:

- New intersectional anatomy experiments to demonstrate that CCol neurons are members of a broad class of Layer 5b projection system with widespread projections in the forebrain and midbrain (Fig. 1a, as suggested by reviewer 1)
- New quantitative immunolabeling of primary cochlear afferent synapses 1 day and 14 days after moderate noise exposure to confirm the classic “synaptopathy” description (Fig. 3e-g, as suggested by reviewers 1 and 3).
- New measurements of daily changes in ABR thresholds and amplitudes using broadband stimuli following noise exposure to provide a more direct comparison with daily changes in corticofugal response gain (Fig. 4c and 5a-b, as suggested by reviewers 1,2).
- New measurements of the extended time course of CCol changes to include daily imaging for 14 days following noise exposure (Fig. 5c-d, as suggested by all reviewers)
- New measurements from a sham-exposed control group (Fig. 2 and Fig 5b-d, as suggested by reviewer 3).

Specific Comments

1. GCaMP was originally described in Tian et al., 2009 to cause abnormal activity once it enters the nucleus of the cell body. While this is less of a problem with GCaMP6, it still does occur and standard practice is to exclude cells with nuclear filling. How do the authors know that the axons they are recording GCaMP signals from are in healthy cells? This is particularly important given the wide range of responses that the authors see across animals in figure 4, which could be influenced by varying levels of GCaMP expression.

>>> One of the sources behind the “wide range of responses” was purely analytic; in the original manuscript, we computed the DF/F for each trial and then averaged. In the revised manuscript, we average the trials before computing the DF/F. The latter approach is standard and has the effect of reducing measurement variability in the trial-averaged signal. To put a number on it, the coefficient in variation in the peak DF/F response amplitudes on the first day of imaging from the five original mice was 0.62 and the improved analysis method reduces the variance from the same trials in the same mice by nearly 50% (to 0.35).

As for the possible contribution of cellular pathology due to nuclear over-filling, the reviewer identifies a challenge for axon imaging because the expression levels at the cell body are not measured concurrently. For this reason, we perfused the mice once widefield axon imaging was complete to confirm that GCaMP was expressed in deep layer neurons of ACtx without evidence of nuclear filling in the vast majority of neurons. Generally, there is no issue with nuclear filling for when imaging is completed within 8 weeks after virus injection, as per our study. A typical example of GCaMP expression in a fixed section of the ACtx is shown in the adjacent figure.

2. What are the expected levels of variability from GCaMP axon imaging with widefield? Is there at least a study that the authors can refer to using this technique to give the reader an idea of how much variability is expected from the experimental preparation used?

>>> Chronic epifluorescence imaging of population GCaMP signals from axons have not been reported previously, to the

best of our knowledge. For comparison, we cite a study that performed epifluorescence imaging of population GCaMP3 signals from descending axons into the olfactory bulb of awake mice (Rothermel and Wachowiak 2014) and another study that imaged GCaMP6 signals under 2-photon excitation of corticocollicular axons in anesthetized mice using the same preparation described here (Barnstedt et al., 2015). Descending cortical projections are known to be modulated by “internal state” variables related to vigilance, arousal, etc., which could introduce sources of variability. Additional sources of variability in the overall response strength between animals would likely include differences in the number of cortical neurons that express GCaMP and differences in the arrangement of large surface vessels relative to the region of the inferior colliculus accessible for imaging.

As for differences in the degree of cortical plasticity arising from damage to cochlear afferent neurons, this is inherently variable as characterized by our lab in several recent studies (Chambers et al., *Neuron* 2016; Resnik and Polley *eLife*, 2017). In part, this is what makes this topic of study so interesting. What are the factors that cause some networks to recover (and often exceed) baseline responsiveness while other networks do not? Of course, we do see clear differences in extent and form of plasticity arising from the relatively subtle hearing loss protocol used here as compared to the more massive nerve damage in the form of bilateral ouabain application (Resnik and Polley, *eLife* 2017). Although the extent of damage with these protocols are very well controlled (as compared to other hearing loss protocols), there is of course some variability in the degree of damage for a given protocol (either moderate noise exposure or ouabain) that we quantify by explicitly counting cochlear afferent synapses (Yuan et al. *JARO* 2013; Chambers et al. *Neuron* 2016), DPOAE and ABR amplitudes (all studies). Importantly, in each of the studies we have performed, we find no correlation with the degree of cochlear afferent nerve damage for a given hearing loss protocol and the degree of cortical plasticity.

This is a feature, not a bug, as any good animal model of central pathophysiology following acute hearing loss would have to show some inter-individual variability. One of the central mysteries of perceptual disorders such as visual snow, Charles Bonnet syndrome, tinnitus, hyperacusis etc is that two individuals exposed to the same sensory insult (explosions, rock concert, glaucoma, macular degeneration etc.) can have very different perceptual outcomes in the form of hallucinations, hypersensitivity and the like. One of the aims of our research is to identify the physiological features present just before or immediately following hearing loss that do predict the eventual degree of plasticity. Our *eLife* paper last year reported that early dynamics in parvalbumin+ cortical GABA neurons provides the best predictor of the eventual degree of recovery and excess auditory gain observed weeks later.

All of this discussion aside, we assume the Reviewer’s first and second questions were motivated by a fundamental concern about the robustness and validity of the main effects reported in the original manuscript. The best response we can provide for this concern comes in the form of new data, not theoretical argument. For this reason, we added a control group and doubled the number of mice in the treatment group as well as the duration of imaging and fixing some simple issues in the data analysis (described above). The end result of these efforts is a much cleaner, convincing data set that we hope will satisfy the reviewer’s concerns.

3. In Xu et al., *Nat. Neurosci*, 2007 and Holtmaat et al., *Nat. Protocols*, 2009, it is shown that there is a strong upregulation of astrocytes and microglia in the first few weeks (2-4 weeks) after surgery. The authors here only wait 5 days between surgery and imaging. Could the authors please confirm that there is not a strong immune response in their preparations, which could contribute to variability in activity levels between animals (perhaps through immunostaining of their surgical preparations after 5 days or by providing control animals which would help separate effects of experimental preparation from effects due to auditory insult).

>>> As the reviewer suggested, we addressed this by performing the same imaging experiment in a new cohort of non-exposed control mice. In the absence of acoustic trauma, the growth slope of the population corticofugal axon response is stable from one day to the next (Fig. 2). This doesn’t rule out an immune response related to our cranial window surgery, but it does provide good evidence that any such changes would be unlikely to explain the response changes we report, as there is no reason to think it would differ between the control and treatment groups. Running the control group seemed like a more direct and informative way to address this point than attempting to quantify changing levels of astrocytes and microglia and inferring whether such levels would affect sound-evoked responses we measure.

4. I do not see that authors have checked whether they actually create a loss of afferent synapses with their protocol. While the histology showing this would be nice, I think that their wording throughout should at least reflect that they have not checked. For example, in the abstract there is really no evidence in this paper that there is a compensation for

a permanent loss of cochlear afferent synapses. Or in line 171-172, it is not clear that they are measuring during a period of permanent loss of cochlear afferent synapses.

>>> The loss of cochlear synapses after moderate noise exposure has been demonstrated in over a dozen papers and can be accurately estimated by quantifying the amplitude of ABR wave 1. Regardless, we performed quantitative cochlear immunolabeling to quantify the synapse loss 1 day and 14 days after noise exposure. As per previous studies, we observed that approximately 50% of the cochlear afferent synapses are lost within 24 hours of moderate noise exposure and this value remains unchanged weeks later. These data are provided in Figure 3e-g.

5. In line 175-177, I think that the statement they make here should be quantified. When I look at figure 4, it is not clear to me that the pattern is repeatable, or that animals 3, 4, or 5 have a stabilization, which to me would mean being at baseline for multiple time points. If they want to say this, they need to quantify what they mean by each part of the statement and provide analyses with statistics to back up their statements. Given the strong variability in their signals, I think that a few control animals would greatly strengthen their ability to draw conclusions from their data (and provide a robust way to do statistics).

>>> Overall, we agree with the reviewer and have performed new experiments to address the points she/he raised. To her/his points:

- 1) in the absence of cochlear synapse damage, the auditory response gain remains stable over the measurement period in five control mice (Fig. 2)
- 2) Within 24 hours following noise exposure the auditory response gain is reduced in 10/10 mice
- 3) Over the following 2-4 days, the response gain is increased above baseline in 10/10 mice
- 4) Per the reviewer's suspicion, we found that enhanced response gain remained elevated when we analyzed the data from a larger sample of noise-exposed mice (from five to ten mice) and doubled the duration of the imaging period from 7 to 14 days.
- 5) Per the reviewer's request, each of the above observations have been appropriately quantified and supported with statistics.

6. What do the authors mean by homeostatic plasticity? Just that activity returns? Typically, recovery of activity that is termed homeostatic is associated with a homeostatic mechanism (like synaptic scaling for example). I do not object to the use of the term without an accompanying homeostatic mechanism, but I just would like to know why do they think this is a homeostatic response instead of a Hebbian or adaptation response - especially give its time course of days? Could they please include this in the discussion?

>>> After carefully considering the reviewer's point, we now avoid using the term "homeostatic plasticity" except on pages 12-13 of the Discussion where we explicitly compare our observations to that literature. The increased gain we report in these excitatory projection neurons *could* be a component of a homeostatic plasticity process that regulates the net activity levels of cortical ensembles (in the same way that increased mEPSC amplitudes allow single neurons to restore baseline excitability levels). But as the reviewer points out, "homeostatic plasticity" really describes a synaptic plasticity that stabilizes the activity level of a neuron. We explicitly discuss how homeostatic or Hebbian mechanisms could support the plasticity reported here and identify future experiments that could be used to reconcile central auditory plasticity with the larger literature on compensatory plasticity in other sensory systems.

7. The authors use the term 'multiplicative scaling' in line 187 and again in the figure legend. What exactly do they mean by that? Given that they are using the term 'homeostatic plasticity', I assume this term to mean that there is a shift of a given distribution by a multiplicative factor (typically miniature EPSCs or IPSCs), such that you take the distribution before deprivation, multiply it by a factor and then it overlays with the after deprivation distribution. It is fine if they want to apply this term to their response curves in figure 3, but then the entire distribution should shift. All that is changing in both figure 3A and B is the 80dB data point, which to me, does not seem like multiplicative scaling. If they want to use this term (although I think it is a pretty loaded term so personally I would use different wording), it would be helpful if they defined it and quantified how they decided their changes were multiplicative.

>>> We agree that it is a loaded term and no longer use the term "multiplicative scaling" in the manuscript. We use the terms "divisive" or "multiplicative" because they describe the mathematical operation that converts the baseline response function to that we observe hours after or days after cochlear synapse damage, respectively. Sure, Figure 3A and B in the last manuscript showed an example mouse where the increased response was most prominent at 80 dB SPL response (but was also observed to a lesser extent at 70 dB SPL). Our goal is to quantify changes in response gain. In any

system, gain describes the increase in output per unit increase in the input. We take the CCol response amplitude measured at baseline and again at various points after cochlear nerve damage. We fit the region of linear growth across the five highest sound levels for a given pair of imaging sessions with a 1st order polynomial and estimate the change in response gain according to the slope of the fit line. This is a standard approach used to quantify changes in cortical response gain after various types of manipulations (e.g., Olsen, Scanziani et al., Nature 2012; Nelson and Mooney, Neuron 2016; Guo, Polley et al., Neuron 2017). The goodness of fit for this linear estimate in our data is quite high ($R^2 = 0.8 \pm 0.03$, on average across animals and imaging sessions), so any changes in the slope are unlikely to arise solely from any one sound level.

8. There is a huge range in the individual mouse responses in figure 4. Is there any correlation with the behavioral measures (DPOAE and ABR) and the GCaMP response of the animal? If not, could they mention in the discussion why they think there is not a correlation between activity and behavior.

>>> As mentioned above, a few simple improvements in the data analysis reduced variability between mice. Doubling the sample size, extending the imaging period and adding a control also lends confidence. The ABR and DPOAE are physiological responses, not behaviors. Regardless, the baseline ABR w1 amplitudes do not predict the degree of multiplicative gain measured 2-4 days following noise exposure ($R^2 = -0.526$; $p = 0.363$). As the reviewer can appreciate in Figs. 4 and 5, the initial suppression and subsequent over-compensation in response gain are quite consistent.

9. Are there significant differences in figure 2d? could they please label them in the figure with asterisks or NS? Please label all statistics in the figure legends and use standard marking on the figures.

>>> Yes, the shifts in ABR threshold (Fig. 3B), DPOAE threshold (Fig. 3C) and ABR amplitude (Fig. 3D) are all significant 24 hours after noise exposure. The decrease in ABR amplitude at high frequencies that correspond to the region of afferent synapse damage is significantly decreased 2 weeks after noise exposure (Fig. 3D). These differences (and lack thereof) have been labeled in Figure 3 (formerly Figure 2) and labeled in the figure legend.

10. In line 218, Could the authors please elaborate about what they consider 'precise' about the homeostatic adjustment? Is it the timing or the degree (or something else)? Additionally, the authors contrast the current study to central gain potentiation, which "over-shoots the mark" (line 220) elevating central excitability. The authors data also over-shoot the mark and it is not clear the mechanism by which that happens, but there is no obvious reason to me to assume that it is not through a change in excitability. Could they please clarify?

>>> This is covered in some detail in the revised Discussion on pg. 12-13, though without unnecessary adjectives such as "precise".

11. In the section of the discussion on Central Gain Enhancement (lines 233-252), it is not clear to me how this section relates to the current data, which is not mentioned in the entirety of this section. I would either explain how this is relevant to the current study, or remove it. Same is true for the next section on Hierarchical Regulation (lines 255-274).

>>> As the same point was raised by Reviewer 2, we removed these sections and instead focus the Discussion on the potential consequences of increased response gain in projection neurons and potential mechanisms that may underlie the changes we report.

12. Again in line 298, only mouse 2 (and maybe 1) really stabilizes at baseline levels (assuming stabilization is multiple time points). So I would remove this emphasis or provide statistics of which mice and timepoints are not statistically significantly different from baseline.

>>> Agreed. This has been removed.

13. In lines 298-302, the authors suggest that their time course fits the signature of a homeostatic process that dips in response, compensates at 24 hours, then overshoots before coming back to baseline. In Hengen et al., the paper the authors cite as the basis for this signature, Hengen et al. specifically state that there is not an overshoot in their data:

"Crucially, over the next 2 days of MD (MD3–MD4), firing rates rebounded and by MD5–MD6 were indistinguishable from baseline. Although mean firing rates were ~9% higher on MD6 (P32) relative to baseline (P26), this increase was within the range of variation in the control hemisphere (Figure 2C) and was not significant ($p = 0.98$, Figure 2D, Tukey-Kramer test)." (Hengen et al., 2013)

This section should be reworded to either include references that support their description or to reflect what is reported in Hengen et al., 2013.

>>> We thank the reviewer for pointing this out to us. We removed this reference in the process of overhauling the Discussion.

14. From the methods, it seems that the authors see the strongest effect 30-60 minutes after their noise exposure. Given that they do not do any behavior measures until at least one day later, how do they know that the animals are actually suffering from hearing loss so quickly? Could they not just have a short-term adaptation, which goes away? What would happen if they record 6 or 12 hours after exposure or if they measured the ABR 30 minutes after the noise exposure? If the animals are not suffering from hearing damage, is the response actually homeostatic?

>>> The damaging effects of acoustic over-exposure on auditory nerve terminals are thought to arise from excitotoxicity occurring during exposure to sustained intense noise, as described in many previous studies. Regardless, the reviewer is correct that we did not provide evidence of peripheral hearing loss during the period when the central response gain is changing. To address this, we performed new experiments that measure the ABR immediately following noise exposure and then again with single day resolution for the following six days. Further, we also performed experiments to quantify the loss of primary afferent synapses from the cochlear nerve onto inner hair cells 24 hours and 2 weeks following noise exposure. Collectively, these new data confirm many prior reports that damage to afferent nerve terminals occurs very rapidly and the loss remains stable for weeks. There are other temporary changes in the sensory cells (most likely arising from stereocilia tip links) that introduce additional changes in ABR thresholds and wave 1 amplitudes immediately after noise exposure but reverse within 2 days. In fact, a 2015 paper observed that the cochlear synaptic damage was at maximum levels immediately after the sound exposure ended (0 hours, Liberman et al. 2015, cited here). Collectively, these new data address the reviewer's point by confirming that the effects of noise exposure on the periphery is rapid and already pronounced at the earliest time points measured.

15. Related to that last point, can the authors rule out the effects of stress as a result of loud noise exposure? For example, are there changes in response to the cerebellum activity levels (as in figure 1) immediately after noise exposure?

>>> We do not rule out that exposure to loud noise can induce stress. However, it is hard to imagine how the effects we report here would arise from elevated stress. We describe changes in neural activity locked to the onset of brief sound stimuli. As the GCaMP is only expressed in the axons of neurons from auditory cortex, there is no signal arising directly from the cerebellum. Also, just to put things in context, the levels of noise used to here to create a temporary threshold shift are considered quite moderate. They are well below the pain threshold and are akin to what would enter the ear canal for a few hours at a typical rock concert.

16. When exactly were the animals injected with ketamine/xylazine and does that affect their GCaMP activity levels?

>>> We measured the ABR under ketamine/xylazine anesthesia immediately following the imaging session 2 days before noise exposure and 1 day after noise exposure. It follows that there would be approximately 23 hours to clear these fast-acting anesthetics prior to the next imaging session. It seems unlikely that any residual effect of the anesthetic would be observed in the following session. This has confirmed by observing that there are no systematic differences in our measurements between the two baseline imaging sessions or days 2, 3 or 4.

17. In the methods, it says that the ABR was done 2 days after noise exposure and in the results and figures, 1 day after exposure. Could the authors clarify this?

>>> This has been clarified in the text. In our first set of experiments, we measured ABR and DPOAE using tone burst of varying frequency before noise exposure, 1 day after noise exposure and again 14 days after noise exposure. In our new experiments, we measure ABR threshold and amplitude with a noise burst stimulus several hours after noise exposure and every day thereafter for the first week.

18. What frequency did the authors use in figure 3A-B?

>>> Corticofugal axon responses were evoked with bursts of white noise (0-50 kHz bandwidth). By also describing changes in ABR wave 1 amplitude with noise burst stimuli, we provide a more direct "apples to apples" basis for contrasting changes in auditory nerve and descending corticofugal axon responsiveness (Fig. 4 and Fig. 5).

REVIEWERS' COMMENTS:

Reviewer #1 (Remarks to the Author):

Review of revision of Asokan et al 2017-2018

The authors thoroughly addressed criticisms of their earlier work by performing a significant number of new experiments (all of figure 1, figure 4C), and repeating experiments to extend timelines throughout the paper (Figure 5B). They also improve explanations and details of analyses throughout. An impressive amount of effort went into addressing reviewer concerns, and the paper is quite improved in both rigor and interpretability. In addition, the new / revised conclusion that CCol response gain remains elevated following noise-induced cochlear damage makes more sense in light of the consistent suppression of ABR wave 1 thresholds.

A couple of (very) minor points remain:

- Figure legend, Figure 2, within 2D: line 690 "response amplitude measured on the day specified (y-axis)." Implies that y-axis = day. Re-write as "response amplitude (y-axis) measured on the day specified."
- Figure 3F: Please find ways to differentiate the various immunolabeled puncta in addition to the color-coded arrows, for color-blind individuals (eg line sizes, arrow heads, etc.).
- Results text line 227 – auditory nerve response gain is not recovered at D1, better wording might be "partially recovered from D1 to D2" or similar.
- Results text line 237 – how is this an "estimate"? It was measured.

Reviewer #2 (Remarks to the Author):

This is a deeply revised version of the study. The authors conducted additional experiments, both in controls and experimental animals, and the new data now indicate that the increased CCol growth functions remains elevated over the two observed weeks instead of returning to baseline as the previous version of the study concluded. The revised conclusion makes more sense and is more consistent with previous results from cortical circuits by this group. Most importantly, the new experiments increase the soundness of the results and conclusions. The authors also have addressed most of my other comments and the re-writing, especially as it concerns the discussion, has improved the manuscript as well.

Unfortunately, the authors did not address any mechanisms of the gain increase, although they provide likely interpretations based on their previous results from auditory cortex, or its effect on the IC, so the study remains a bit descriptive, yet tantalizing. While the demonstration of a gain increase in a descending projection is novel, the changes observed in the CCol projection now largely fit in the general picture of a hearing-loss induced gain increase that we obtained from numerous studies of ascending pathways.

The authors now include new tracing experiments from 2 mice in which they show that CCol axons also send collaterals to the striatum and lateral amygdala. This suggests that the increased gain they observed in the IC extends to these areas as well, which, as they

discussed, is quite interesting in respect to the non-auditory components that are associated with hearing loss and tinnitus. I do not know how well the striatum and lateral amygdala can be identified in sections stained with DAPI, which usually does not delineate nuclear boundaries very well. At a minimum an overview photo of the termination areas in the lateral amygdala and posterior regions of the striatum should be included in Fig. 1.

Reviewer #3 (Remarks to the Author):

I am happy with the authors' revisions and believe that they have substantially improved the manuscript. I believe that it is now appropriate for publication in Nature Communications.

Reviewer #1 (Remarks to the Author):

The authors thoroughly addressed criticisms of their earlier work by performing a significant number of new experiments (all of figure 1, figure 4C), and repeating experiments to extend timelines throughout the paper (Figure 5B). They also improve explanations and details of analyses throughout. An impressive amount of effort went into addressing reviewer concerns, and the paper is quite improved in both rigor and interpretability. In addition, the new / revised conclusion that CCol response gain remains elevated following noise-induced cochlear damage makes more sense in light of the consistent suppression of ABR wave 1 thresholds.

A couple of (very) minor points remain:

- Figure legend, Figure 2, within 2D: line 690 “response amplitude measured on the day specified (y-axis).” Implies that y-axis = day. Re-write as “response amplitude (y-axis) measured on the day specified.”

>>> **Done**

- Figure 3F: Please find ways to differentiate the various immunolabeled puncta in addition to the color-coded arrows, for color-blind individuals (eg line sizes, arrow heads, etc.).

>>> **The arrow style has been modified per the reviewer’s request.**

- Results text line 227 – auditory nerve response gain is not recovered at D1, better wording might be “partially recovered from D1 to D2” or similar.

- Results text line 237 – how is this an “estimate”? It was measured.

>>> **“estimated” has now been replaced with “measured”.**

Reviewer #2 (Remarks to the Author):

This is a deeply revised version of the study. The authors conducted additional experiments, both in controls and experimental animals, and the new data now indicate that the increased of CCol growth functions remains elevated over the two observed weeks instead of returning to baseline as the previous version of the study concluded. The revised conclusion makes more sense and is more consistent with previous results from cortical circuits by this group. Most importantly, the new experiments increase the soundness of the results and conclusions. The authors also have addressed most of my other comments and the re-writing, especially as it concerns the discussion, has improved the manuscript as well.

Unfortunately, the authors did not address any mechanisms of the gain increase, although they provide likely interpretations based on their previous results from auditory cortex, or its effect on the IC, so the study remains a bit descriptive, yet tantalizing. While the demonstration of a gain increase in a descending projection is novel, the changes observed in the CCol projection now largely fit in the general picture of a hearing-loss induced gain increase that we obtained from numerous studies of ascending pathways.

The authors now include new tracing experiments from 2 mice in which they show that CCol axons also send collaterals to the striatum and lateral amygdala. This suggests that the increased gain they observed in the IC extends to these areas as well, which, as they discussed, is quite interesting in respect to the non-auditory components that are associated with hearing loss and tinnitus.

I do not know how well the striatum and lateral amygdala can be identified in sections stained with DAPI, which usually does not delineate nuclear boundaries very well. At a minimum an overview photo of the termination areas in the lateral amygdala and posterior regions of the striatum should be included in Fig. 1.

>>> The reviewer is correct that DAPI does not delineate nuclear boundaries very well. However, the striatum can be easily identified using the lateral margin of the lateral ventricle. The boundary between the striatum and lateral amygdala is formed by the external capsule, which can also be easily visualized by the fluorescent fiber bundle. We now include an overview photo in Fig.1 that shows both of these landmarks, per the reviewer's request.